# Structured Denoising Diffusion Models in Discrete State-Spaces

**Jacob Austin,**[*] **Daniel D. Johnson,**[*] **Jonathan Ho, Daniel Tarlow & Rianne van den Berg**[†]
Google Research, Brain Team
{jaaustin,ddjohnson,jonathanho,dtarlow,riannevdberg}@google.com

## Abstract

Denoising diffusion probabilistic models (DDPMs) [17] have shown impressive results on image and waveform generation in continuous state spaces. Here, we introduce Discrete Denoising Diffusion Probabilistic Models (D3PMs), diffusion-like generative models for discrete data that generalize the multinomial diffusion model of Hoogeboom et al. [18], by going beyond corruption processes with uniform transition probabilities. This includes corruption with transition matrices that mimic Gaussian kernels in continuous space, matrices based on nearest neighbors in embedding space, and matrices that introduce absorbing states. The third allows us to draw a connection between diffusion models and autoregressive and mask-based generative models. We show that the choice of transition matrix is an important design decision that leads to improved results in image and text domains. We also introduce a new loss function that combines the variational lower bound with an auxiliary cross entropy loss. For text, this model class achieves strong results on character-level text generation while scaling to large vocabularies on LM1B. On the image dataset CIFAR-10, our models approach the sample quality and exceed the log-likelihood of the continuous-space DDPM model.

## 1   Introduction

Generative modeling is a core problem in machine learning, useful both for benchmarking our ability to capture statistics of natural datasets and for downstream applications that require generating high-dimensional data like images, text, and speech waveforms. There has been a great deal of progress with the development of methods like GANs [14, 3], VAEs [22, 32], large autoregressive neural network models [43, 42, 44], normalizing flows [31, 11, 21, 30], and others, each with their own tradeoffs in terms of sample quality, sampling speed, log-likelihoods, and training stability.

Recently, diffusion models [36] have emerged as a compelling alternative for image [17, 39] and audio [7, 23] generation, achieving comparable sample quality to GANs and log-likelihoods comparable to autoregressive models with fewer inference steps. A diffusion model is a parameterized Markov chain trained to reverse a predefined forward process, which is a stochastic process constructed to gradually corrupt training data into pure noise. Diffusion models are trained using a stable objective closely related to both maximum likelihood and score matching [19, 45], and they admit faster sampling than autoregressive models by using parallel iterative refinement [28, 38, 40, 37].

Although diffusion models have been proposed in both discrete and continuous state spaces [36], most recent work has focused on Gaussian diffusion processes that operate in continuous state spaces (e.g. for real-valued image and waveform data). Diffusion models with discrete state spaces have

35th Conference on Neural Information Processing Systems (NeurIPS 2021).

---

[*]Equal contributions
[†]Now at Microsoft Research

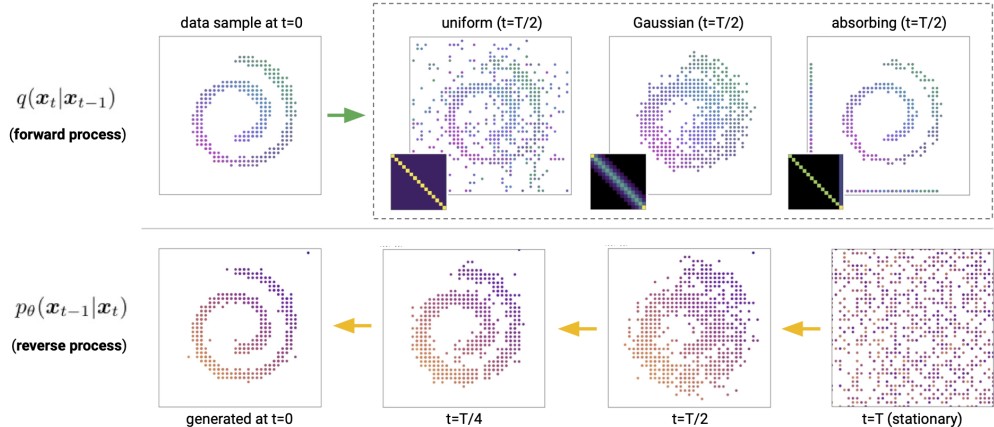

Figure 1: D3PM forward and (learned) reverse process applied to a quantized swiss roll. Each dot represents a 2D categorical variable. Top: samples from the uniform, discretized Gaussian, and absorbing state D3PM model forward processes, along with corresponding transition matrices $\boldsymbol{Q}$. Bottom: samples from a learned discretized Gaussian reverse process.

been explored for text and image segmentation domains [18], but they have not yet been demonstrated as a competitive model class for large scale text or image generation.

Our aim in this work is to improve and extend discrete diffusion models by using a more structured categorical corruption process to shape data generation, as illustrated in Figure 1. Our models do not require relaxing or embedding discrete data (including images) into continuous spaces, and can embed structure or domain knowledge into the transition matrices used by the forward process. We achieve significantly improved results by taking advantage of this flexibility. We develop structured corruption processes appropriate for text data, using similarity between tokens to enable gradual corruption and denoising. Expanding further, we also explore corruption processes that insert [MASK] tokens, which let us draw parallels to autoregressive and mask-based generative models. Finally, we study discrete diffusion models for quantized images, taking inspiration from the locality exploited by continuous diffusion models. This leads to a particular choice of discrete corruption process that diffuses preferentially to more similar states and leads to much better results in the image domain.

Overall, we make a number of technical and conceptual contributions. Beyond designing several new structured diffusion models, we introduce a new auxiliary loss which stabilizes training of D3PMs and a family of noise schedules based on mutual information that lead to improved performance. We strongly outperform various non-autoregressive baselines for text generation on character-level text generation, and successfully scale discrete diffusion models to large vocabularies and long sequence lengths. We also achieve strong results on the image dataset CIFAR-10, approaching or exceeding the Gaussian diffusion model from Ho et al. [17] on log-likelihoods and sample quality.

## 2   Background: diffusion models

Diffusion models [36] are latent variable generative models characterized by a forward and a reverse Markov process. The forward process $q(\boldsymbol{x}_{1:T}|\boldsymbol{x}_0) = \prod_{t=1}^{T} q(\boldsymbol{x}_t|\boldsymbol{x}_{t-1})$ corrupts the data $\boldsymbol{x}_0 \sim q(\boldsymbol{x}_0)$ into a sequence of increasingly noisy latent variables $\boldsymbol{x}_{1:T} = \boldsymbol{x}_1, \boldsymbol{x}_2, ..., \boldsymbol{x}_T$. The learned reverse Markov process $p_\theta(\boldsymbol{x}_{0:T}) = p(\boldsymbol{x}_T) \prod_{t=1}^{T} p_\theta(\boldsymbol{x}_{t-1}|\boldsymbol{x}_t)$ gradually denoises the latent variables towards the data distribution. For example, for continuous data, the forward process typically adds Gaussian noise, which the reverse process learns to remove.

In order to optimize the generative model $p_\theta(\boldsymbol{x}_0)$ to fit the data distribution $q(\boldsymbol{x}_0)$, we typically optimize a variational upper bound on the negative log-likelihood:

$$L_{\text{vb}} = \mathbb{E}_{q(\boldsymbol{x}_0)}\bigg[\underbrace{D_{\text{KL}}[q(\boldsymbol{x}_T|\boldsymbol{x}_0)||p(\boldsymbol{x}_T)]}_{L_T} + \sum_{t=2}^{T}\underbrace{\mathbb{E}_{q(\boldsymbol{x}_t|\boldsymbol{x}_0)}\big[D_{\text{KL}}[q(\boldsymbol{x}_{t-1}|\boldsymbol{x}_t,\boldsymbol{x}_0)||p_\theta(\boldsymbol{x}_{t-1}|\boldsymbol{x}_t)]\big]}_{L_{t-1}}$$
$$\underbrace{-\mathbb{E}_{q(\boldsymbol{x}_1|\boldsymbol{x}_0)}[\log p_\theta(\boldsymbol{x}_0|\boldsymbol{x}_1)]}_{L_0}\bigg]. \tag{1}$$

When the number of time steps $T$ goes to infinity, both the forward process and the reverse process share the same functional form [36, 12], in the sense that the true posterior $q(\boldsymbol{x}_{t-1}|\boldsymbol{x}_t)$ becomes fully conditionally independent.[3] This motivates using a conditionally independent approximate reverse process $p_\theta(\boldsymbol{x}_{t-1}|\boldsymbol{x}_t)$ from the same class of distributions as that of the forward process. Furthermore, for several choices of the forward process the distribution $q(\boldsymbol{x}_t|\boldsymbol{x}_0)$ converges to a stationary distribution $\pi(\boldsymbol{x})$ in the limit $t \to \infty$ independent of the value of $\boldsymbol{x}_0$. When the number of time steps $T$ is large enough and we choose $\pi(\boldsymbol{x})$ as the prior $p(\boldsymbol{x}_T)$, we can guarantee that the $L_T$ term in (1) will approach zero regardless of the data distribution $q(\boldsymbol{x}_0)$. (Alternatively, one can use a learned prior $p_\theta(\boldsymbol{x}_T)$.)

While $q(\boldsymbol{x}_t|\boldsymbol{x}_{t-1})$ can in theory be arbitrary, efficient training of $p_\theta$ is possible when $q(\boldsymbol{x}_t|\boldsymbol{x}_{t-1})$:

1. Permits efficient sampling of $\boldsymbol{x}_t$ from $q(\boldsymbol{x}_t|\boldsymbol{x}_0)$ for an arbitrary time $t$, allowing us to randomly sample timesteps and optimize each $L_{t-1}$ term individually with stochastic gradient descent,

2. Has a tractable expression for the forward process posterior $q(\boldsymbol{x}_{t-1}|\boldsymbol{x}_t,\boldsymbol{x}_0)$, which allows us to compute the KL divergences present in the $L_{t-1}$ term of (1).

The majority of recent work in continuous spaces [17, 37, 7, 28] defines the forward and reverse distributions as $q(\boldsymbol{x}_t|\boldsymbol{x}_{t-1}) = \mathcal{N}\big(\boldsymbol{x}_t|\sqrt{1-\beta_t}\boldsymbol{x}_{t-1},\beta_t\boldsymbol{I}\big)$ and $p_\theta(\boldsymbol{x}_{t-1}|\boldsymbol{x}_t) = \mathcal{N}\big(\boldsymbol{x}_{t-1}|\boldsymbol{\mu}_\theta(\boldsymbol{x}_t,t),\boldsymbol{\Sigma}_\theta(\boldsymbol{x}_t,t)\big)$, respectively. The aforementioned properties hold in the case of these Gaussian diffusion models: the forward process $q(\boldsymbol{x}_t|\boldsymbol{x}_0)$ converges to a stationary distribution, motivating the choice $p(\boldsymbol{x}_T) = \mathcal{N}\big(\boldsymbol{x}_T|\boldsymbol{0},\boldsymbol{I}\big)$, and both $q(\boldsymbol{x}_t|\boldsymbol{x}_0)$ and $q(\boldsymbol{x}_{t-1}|\boldsymbol{x}_t,\boldsymbol{x}_0)$ are tractable Gaussian distributions for which the KL divergence can be computed analytically.

## 3  Diffusion models for discrete state spaces

Diffusion models with discrete state spaces were first introduced by Sohl-Dickstein et al. [36], who considered a diffusion process over binary random variables. Hoogeboom et al. [18] extended the model class to categorical random variables with transition matrices characterized by uniform transition probabilities. In their supplementary material, Song et al. [37] also derived this extension, although no experiments were performed with this model class. Here, we briefly describe a more general framework for diffusion with categorical random variables which includes these models as special cases.[4]

For scalar discrete random variables with $K$ categories $x_t, x_{t-1} \in 1, ..., K$ the forward transition probabilities can be represented by matrices: $[\boldsymbol{Q}_t]_{ij} = q(x_t = j|x_{t-1} = i)$. Denoting the one-hot version of $x$ with the row vector $\boldsymbol{x}$, we can write

$$q(\boldsymbol{x}_t|\boldsymbol{x}_{t-1}) = \text{Cat}(\boldsymbol{x}_t; \boldsymbol{p} = \boldsymbol{x}_{t-1}\boldsymbol{Q}_t), \tag{2}$$

where $\text{Cat}(\boldsymbol{x}; \boldsymbol{p})$ is a categorical distribution over the one-hot row vector $\boldsymbol{x}$ with probabilities given by the row vector $\boldsymbol{p}$, and $\boldsymbol{x}_{t-1}\boldsymbol{Q}_t$ is to be understood as a row vector-matrix product. We assume that $\boldsymbol{Q}_t$ is applied to each pixel of an image or each token in a sequence independently, and that $q$ factorizes over these higher dimensions as well; we thus write $q(\boldsymbol{x}_t|\boldsymbol{x}_{t-1})$ in terms of a single

---

[3]For continuous state spaces and Gaussian $q$, the limit $T \to \infty$ corresponds to a stochastic differential equation [40], whereas for discrete state spaces it corresponds to a Markov jump process.

[4]Our implementation of D3PM framework is available at `https://github.com/google-research/google-research/tree/master/d3pm`.

element. Starting from $\boldsymbol{x}_0$, we obtain the following $t$-step marginal and posterior at time $t-1$:

$$q(\boldsymbol{x}_t|\boldsymbol{x}_0) = \text{Cat}\left(\boldsymbol{x}_t; \boldsymbol{p} = \boldsymbol{x}_0\overline{\boldsymbol{Q}}_t\right), \quad \text{with} \quad \overline{\boldsymbol{Q}}_t = \boldsymbol{Q}_1\boldsymbol{Q}_2\ldots\boldsymbol{Q}_t$$

$$q(\boldsymbol{x}_{t-1}|\boldsymbol{x}_t, \boldsymbol{x}_0) = \frac{q(\boldsymbol{x}_t|\boldsymbol{x}_{t-1}, \boldsymbol{x}_0)q(\boldsymbol{x}_{t-1}|\boldsymbol{x}_0)}{q(\boldsymbol{x}_t|\boldsymbol{x}_0)} = \text{Cat}\left(\boldsymbol{x}_{t-1}; \boldsymbol{p} = \frac{\boldsymbol{x}_t\boldsymbol{Q}_t^\top \odot \boldsymbol{x}_0\overline{\boldsymbol{Q}}_{t-1}}{\boldsymbol{x}_0\overline{\boldsymbol{Q}}_t\boldsymbol{x}_t^\top}\right). \quad (3)$$

Note that due to the Markov property of the forward process $q(\boldsymbol{x}_t|\boldsymbol{x}_{t-1}, \boldsymbol{x}_0) = q(\boldsymbol{x}_t|\boldsymbol{x}_{t-1})$. Assuming that the reverse process $p_\theta(\boldsymbol{x}_t|\boldsymbol{x}_{t-1})$ is also factorized as conditionally independent over the image or sequence elements, the KL divergence between $q$ and $p_\theta$ can be computed by simply summing over all possible values of each random variable; we thus satisfy criteria 1 and 2 discussed in Section 2. Depending on $\boldsymbol{Q}_t$, the cumulative products $\overline{\boldsymbol{Q}}_t$ can often be computed in closed form, or simply precomputed for all $t$. However, for large $K$ and large $T$ this may be prohibitive. In Appendix A.4 we discuss how to ensure $\overline{\boldsymbol{Q}}_t$ can still be computed efficiently in this case, allowing the framework to scale to a larger number of categories.

In the next section we discuss the choice of the Markov transition matrices $\boldsymbol{Q}_t$ and corresponding stationary distributions. From here on, we refer to the general class of diffusion models with discrete state spaces as Discrete Denoising Diffusion Probabilistic Models (D3PMs).

## 3.1 Choice of Markov transition matrices for the forward process

An advantage of the D3PM framework described above is the ability to control the data corruption and denoising process by choosing $\boldsymbol{Q}_t$, in notable contrast to continuous diffusion, for which only additive Gaussian noise has received significant attention. Besides the constraint that the rows of $\boldsymbol{Q}_t$ must sum to one to conserve probability mass, the only other constraint in choosing $\boldsymbol{Q}_t$ is that the rows of $\overline{\boldsymbol{Q}}_t = \boldsymbol{Q}_1\boldsymbol{Q}_2\ldots\boldsymbol{Q}_t$ must converge to a known stationary distribution[5] when $t$ becomes large, which can be guaranteed while imposing minimal restrictions on $\boldsymbol{Q}_t$ (see Appendix A.1).

We argue that for most real-world discrete data, including images and text, it makes sense to add domain-dependent structure to the transition matrices $\boldsymbol{Q}_t$ as a way of controlling the forward corruption process and the learnable reverse denoising process. Below we briefly discuss the uniform transition matrices that have been studied in prior work [18], along with a set of structured transition matrices we have explored for our image and text dataset experiments; see Appendix A.2 for more details on each matrix type. We also note that this set is not exhaustive, and many other transition matrices could also be used within the D3PM framework.

**Uniform (Appendix A.2.1).** Sohl-Dickstein et al. [36] considered a simple $2 \times 2$ transition matrix for binary random variables. Hoogeboom et al. [18] later extended this to categorical variables, proposing a transition matrix $\boldsymbol{Q}_t = (1 - \beta_t)\boldsymbol{I} + \beta_t/K\ \mathbb{1}\mathbb{1}^T$ with $\beta_t \in [0, 1]$. Since this transition matrix is doubly stochastic with strictly positive entries, the stationary distribution is uniform. Because the transition probability to any other state is uniform, in this paper we equivalently refer to this discrete diffusion instance as D3PM-uniform.

**Absorbing state (Appendix A.2.2).** Motivated by the success of BERT [10] and recent work on Conditional Masked Language Models (CMLMs) in text, we consider a transition matrix with an absorbing state (called [MASK]), such that each token either stays the same or transitions to [MASK] with some probability $\beta_t$. This does not impose particular relationships between categories, similar to uniform diffusion, but still allows corrupted tokens to be distinguished from original ones. Moreover, the stationary distribution is not uniform but has all the mass on the [MASK] token. For images, we reuse the grey pixel as the [MASK] absorbing token.

**Discretized Gaussian (Appendix A.2.3).** Instead of transitioning uniformly to any other state, for ordinal data we propose imitating a continuous space diffusion model by using a discretized, truncated Gaussian distribution. We choose a normalization such that the transition matrix is doubly stochastic, leading to a uniform stationary distribution. This transition matrix will transition between more similar states with higher probability, and is well suited for quantized ordinal data such as images.

**Token embedding distance (Appendix A.2.4).** Textual data does not have ordinal structure, but there may still be interesting semantic relationships. For instance, in a word-level vocabulary

---

[5]If a stationary distribution is not known, we can introduce a learned prior $p_\theta(\boldsymbol{x}_T)$; we note that this is equivalent to extending the forward process by appending a rank-one matrix $\boldsymbol{Q}_{T+1}$ that ignores $\boldsymbol{x}_T$ and produces a deterministic $\boldsymbol{x}_{T+1}$, then learning the reverse step $p_\theta(\boldsymbol{x}_T|\boldsymbol{x}_{T+1}) = p_\theta(\boldsymbol{x}_T)$.

synonyms or closely related words (like "dog" or "cat") may be more similar than other tokens. As a demonstration of the generality of the D3PM framework, we explore using similarity in word embedding space to guide the forward process, and construct a doubly-stochastic transition matrix that transitions more frequently between tokens that have similar embeddings while maintaining a uniform stationary distribution.

For uniform and absorbing-state diffusion, the cumulative products $\overline{\boldsymbol{Q}}_t$ can be computed in closed form (see Appendix A.4.1); the remainder can be precomputed.

## 3.2 Noise schedules

We consider several different options for the noise schedule of the forward process. For discretized Gaussian diffusion, we explore linearly increasing the variance of the Gaussian before discretizing it. (Note that a linear schedule for $\boldsymbol{Q}_t$ leads to a nonlinear amount of cumulative noise in $\overline{\boldsymbol{Q}}_t$.) For uniform diffusion we use the cosine schedule which sets the cumulative probability of a transition to a cosine function, as introduced by Nichol and Dhariwal [28] and adapted by Hoogeboom et al. [18]. For a general set of transition matrices $\boldsymbol{Q}_t$ (such as the one based on token embeddings), previously proposed schedules may not be directly applicable. We consider linearly interpolating the *mutual information* between $\boldsymbol{x}_t$ and $\boldsymbol{x}_0$ to zero, i.e. $I(\boldsymbol{x}_t; \boldsymbol{x}_0) \approx (1 - \frac{t}{T}) H(\boldsymbol{x}_0)$. Interestingly, for the specific case of absorbing-state D3PMs, this schedule reduces to exactly the $(T - t + 1)^{-1}$ schedule proposed by Sohl-Dickstein et al. [36] for a Bernoulli diffusion process. See Appendix A.7 for more details.

## 3.3 Parameterization of the reverse process

While it is possible to directly predict the logits of $p_\theta(\boldsymbol{x}_{t-1}|\boldsymbol{x}_t)$ using a neural network $\mathrm{nn}_\theta(\boldsymbol{x}_t)$, we follow Ho et al. [17] and Hoogeboom et al. [18] and focus on using a neural network $\mathrm{nn}_\theta(\boldsymbol{x}_t)$ to predict the logits of a distribution $\widetilde{p}_\theta(\widetilde{\boldsymbol{x}}_0|\boldsymbol{x}_t)$, which we combine with $q(\boldsymbol{x}_{t-1}|\boldsymbol{x}_t, \boldsymbol{x}_0)$ and a summation over one-hot representations of $\boldsymbol{x}_0$ to obtain the following parameterization

$$p_\theta(\boldsymbol{x}_{t-1}|\boldsymbol{x}_t) \propto \sum_{\widetilde{\boldsymbol{x}}_0} q(\boldsymbol{x}_{t-1}, \boldsymbol{x}_t|\widetilde{\boldsymbol{x}}_0)\widetilde{p}_\theta(\widetilde{\boldsymbol{x}}_0|\boldsymbol{x}_t). \tag{4}$$

We note that under this $\boldsymbol{x}_0$-parameterization the KL divergence $D_{\mathrm{KL}}[q(\boldsymbol{x}_{t-1}|\boldsymbol{x}_t, \boldsymbol{x}_0)||p_\theta(\boldsymbol{x}_{t-1}|\boldsymbol{x}_t)]$ will be zero if $\widetilde{p}_\theta(\widetilde{\boldsymbol{x}}_0|\boldsymbol{x}_t)$ places all of its probability mass on the original value $\boldsymbol{x}_0$. The decomposition of $q(\boldsymbol{x}_{t-1}|\boldsymbol{x}_t, \boldsymbol{x}_0)$ in (3) also provides us with a motivation for this parameterization. According to (3), in a given state $\boldsymbol{x}_t$, the optimal reverse process only takes into account transitions to states for which $q(\boldsymbol{x}_t|\boldsymbol{x}_{t-1})$ is non-zero. Therefore, the sparsity pattern of $\boldsymbol{Q}_t$ determines the sparsity pattern of the ideal reverse transition probabilities in $p_\theta(\boldsymbol{x}_{t-1}|\boldsymbol{x}_t)$. The parameterization in (4) automatically ensures that the learned reverse probability distribution $p_\theta(\boldsymbol{x}_{t-1}|\boldsymbol{x}_t)$ has the correct sparsity pattern dictated by the choice of the Markov transition matrix $\boldsymbol{Q}_t$. This parameterization also lets us perform inference with $k$ steps at a time, by predicting $p_\theta(\boldsymbol{x}_{t-k}|\boldsymbol{x}_t) = \sum q(\boldsymbol{x}_{t-k}, \boldsymbol{x}_t|\widetilde{\boldsymbol{x}}_0)\widetilde{p}_\theta(\widetilde{\boldsymbol{x}}_0|\boldsymbol{x}_t)$.

Finally, when modeling ordinal discrete data, instead of predicting the logits of $\widetilde{p}_\theta(\widetilde{\boldsymbol{x}}_0|\boldsymbol{x}_t)$ directly with the output of a neural net, another option is to model the probabilities with a truncated discretized logistic distribution (see Appendix A.8). This provides an extra ordinal inductive bias to the reverse model and boosts FID and log-likelihood scores for images.

## 3.4 Loss function

While the original diffusion models introduced by Sohl-Dickstein et al. [36] were optimized with the negative variational lower bound $L_{\mathrm{vb}}$ of (1), more recent diffusion models are optimized with different objectives. For instance, Ho et al. [17] derive a simplified loss function ($L_{\mathrm{simple}}$) that reweights the negative variational bound, and Nichol and Dhariwal [28] explore a hybrid loss $L_{\mathrm{hybrid}} = L_{\mathrm{simple}} + \lambda L_{\mathrm{vb}}$ (using one term to learn the predicted mean and the other to learn predicted variance). Inspired by this recent work, we introduce an auxiliary denoising objective for the $\boldsymbol{x}_0$-parameterization of the reverse process, which encourages good predictions of the data $\boldsymbol{x}_0$ at each time step. We combine this with the negative variational lower bound, yielding the following alternative loss function:

$$L_\lambda = L_{\mathrm{vb}} + \lambda\, \mathbb{E}_{q(\boldsymbol{x}_0)}\mathbb{E}_{q(\boldsymbol{x}_t|\boldsymbol{x}_0)}[-\log \widetilde{p}_\theta(\boldsymbol{x}_0|\boldsymbol{x}_t)]. \tag{5}$$

We note that the auxiliary loss resembles the cross entropy term $L_0$ in (1) at $t = 1$, and so one might expect that it is a KL reweighting similar to the one described by Ho et al. [17]. However, our $L_\lambda$ directly supervises the model output $\widetilde{p}_\theta(\widetilde{x}_0|x_t)$. This is in general a stronger source of supervision than any reweighting of the terms in the lower bound (1), which only provides supervision through the sum in (4). To see this, note that for a fixed $x_0$, both $D_{\mathrm{KL}}[q(x_{t-1}|x_t, x_0)||p_\theta(x_{t-1}|x_t)]$ and $E_{q(x_t|x_0)}[-\log \widetilde{p}_\theta(x_0|x_t)]$ are minimized when $\widetilde{p}_\theta(\widetilde{x}_0|x_t)$ has all its mass on the datapoint $x_0$, but for some choices of $q$ there may be a different setting $\widetilde{x}_0 \neq x_0$ that induces the same distribution $p_\theta(x_{t-1}|x_t)$. We find that training with this loss leads to improved quality of image samples.

# 4 Connection to existing probabilistic models for text

In this section we expand on interesting connections between the D3PM framework and several existing probabilistic and language modeling approaches.

**BERT is a one-step diffusion model:** One possible D3PM transition matrix is a combination of a uniform transition matrix and an absorbing state at the [MASK] token (i.e. $Q = \alpha \mathbb{1}e_m^T + \beta \mathbb{1}\mathbb{1}^T/K + (1 - \alpha - \beta)I$, where $e_m$ is a one-hot vector on the [MASK] token). For a one-step diffusion process in which $q(x_1|x_0)$ replaces 10% of tokens with [MASK] and 5% uniformly at random, this leads precisely to the BERT denoising objective, i.e. $L_{vb} - L_T = -\mathbb{E}_{q(x_1|x_0)}[\log p_\theta(x_0|x_1)] = L_{BERT}$, since $L_T$ is a constant independent of $\theta$ (assuming a fixed prior).

**Autoregressive models are (discrete) diffusion models:** Consider a diffusion process that deterministically masks tokens one-by-one in a sequence of length $N = T$: $q([x_t]_i \mid x_0) = [x_0]_i$ if $i < N - t$ else [MASK] . This is a deterministic forward process, so $q(x_{t-1}|x_t, x_0)$ is a delta distribution on the $x_t$ sequence with one fewer mask: $q([x_{t-1}]_i \mid x_t, x_0) = \delta_{[x_t]_i}$ if $i \neq T - t$ else $\delta_{[x_0]_i}$. While this process is not applied independently to each token, it can be recast as an independently-applied diffusion process on the product space $[0...N] \times \mathcal{V}$, where each token is tagged with its position in the sequence, $\mathcal{V}$ is the vocabulary, and $Q$ is an $N \times |\mathcal{V}| \times N \times |\mathcal{V}|$ sparse matrix.

Because all tokens except the one at position $i = T - t$ have deterministic posteriors, the KL divergence $D_{KL}(q([x_{t-1}]_j|x_t, x_0) \mid\mid p_\theta([x_{t-1}]_j|x_t))$ is zero for all other positions. The only token for which this is not true is the token at position $i$, for which $D_{KL}(q([x_{t-1}]_i|x_t, x_0) \mid\mid p_\theta([x_{t-1}]_i|x_t)) = -\log p_\theta([x_0]_i|x_t)$, the standard cross entropy loss for an autoregressive model.

**(Generative) Masked Language-Models (MLMs) are diffusion models:** Generative Masked Language Models ([13], [47]) are generative models that generate text from a sequence of [MASK] tokens. They are usually trained by sampling a sequence $x_0$, masking $k$ tokens according to some schedule, and learning to predict the masked tokens given context. It turns out that a D3PM absorbing ([MASK]) model trained on the usual ELBO objective with the $x_0$-parameterization from 3.3 reduces to a reweighted version of this MLM objective (see Appendix A.3 for a detailed derivation).

# 5 Text generation

For text, we experiment with generation on two datasets: text8 [26], a character-level dataset extracted from English-language Wikipedia, and the One Billion Word dataset (LM1B) [6], a large dataset of shuffled English-language sentences. For both, we train a D3PM uniform model based on the work by Hoogeboom et al. [18] (D3PM uniform) and a model that masks tokens (D3PM absorbing). We also consider a model that transitions uniformly to nearest neighbors in a token embedding space (D3PM NN). We follow Hoogeboom et al. [18] and use $T = 1000$ timesteps, although we are also able to evaluate on fewer due to the parameterization in Section 3.3.

## 5.1 Character-level generation on text8

text8 is a character-level text dataset consisting of a small vocabulary of 27 tokens: the letters 'a'-'z' and the '_' whitespace token. We follow the convention of training and evaluating text8 in chunks of length 256 without any preprocessing [18]. For nearest-neighbor D3PM, our nearest neighbor graph in character-space is shown in Appendix B.2.1. D3PM uniform models were trained with a cosine schedule from Hoogeboom et al. [18] (ablations in Appendix B.2.1), while D3PM absorbing and D3PM NN models were trained with a mutual information schedule.

Table 1 shows that for D3PM, the D3PM absorbing model performed the best, exceeding the uniform and NN diffusion models. We were able to improve upon the baseline result of [18] with hyperparameter tuning, and our uniform and NN results outperformed results from Hoogeboom et al. [18] across all inference steps, down to as few as 20. We found that $L_{\lambda=0.01}$ worked best for D3PM absorbing, while $L_{vb}$ was better for D3PM uniform. Our model outperforms all non-autoregressive baselines except one, the Discrete Flow model [41] (for which unfortunately no open-source implementations exist), and is also faster than all but one method, the IAF/SCF model [49]. It is also nearly 20x faster than an autoregressive transformer of the same size. We note that while our 20-step D3PM models in Table 1 are much faster than a comparable autoregressive transformers, this table only shows timings for batch size 1 (per device). For larger batches, autoregressive caching allows transformers to perform inference relatively more quickly. We include additional benchmarks and a plot of inference time as a function of iterations in Appendix B.2.1. D3PM with the mask absorbing token was by far the best performing model, which lends credibility to the use of masks in denoising auto-encoders. Nearest-neighbor diffusion only narrowly improves upon a D3PM-uniform model: this was a surprising negative result for us, suggesting that not all notions of structure are meaningful.

## 5.2 Text generation on LM1B

Text generation for large-scale text datasets and large vocabularies with discrete diffusion models has not been previously demonstrated. We include results from LM1B as a proof of concept, showing that these models can indeed scale (as discussed in Appendix A.4), and that the D3PM absorbing model continues to excel. All models were trained and evaluated on packed sequences of length 128, using a sentencepiece[6] vocabulary of size 8192.

Table 2 contains results from experiments on LM1B. Overall, mask diffusion (D3PM absorbing) does relatively well, approaching the performance of a comparable autoregressive model of the same size, and scaling to far fewer steps, while uniform diffusion performs significantly worse. We find, surprisingly, that the D3PM NN model performs worse than the uniform model in terms of log likelihoods (although it demonstrates unique qualitative behavior). This suggests that word embedding similarity may not be a meaningful kind of locality in a diffusion process. We found the the $L_{\lambda=0.01}$ loss worked best for the mask absorbing model, but reduced performance for the other models. We note the surprising scaling in perplexity in Figure 2, achieving strong results with as few as 10 inference steps. We also show samples from our model and completions from corrupted samples.

Table 1: Quantitative results on text8. NLL is reported on the entire test set. Sample times are for generating a single example of length 256. Results are reported on two seeds. All models are standard 12-layer transformers unless otherwise noted. [†]Transformer XL is a 24-layer transformer, using a 784 context window. [‡]Results reported by [18] by running code from official repository.

| Model | Model steps | NLL (bits/char) ($\downarrow$) | Sample time (s) ($\downarrow$) |
| --- | --- | --- | --- |
| Discrete Flow [41] ($8 \times 3$ layers) | - | 1.23 | 0.16 |
| Argmax Coupling Flow [18] | - | 1.80 | $0.40 \pm 0.03$ |
| IAF / SCF [49][‡] | - | 1.88 | $0.04 \pm 0.0004$ |
| Multinomial Diffusion (D3PM uniform) [18] | 1000 | $\leq 1.72$ | $26.6 \pm 2.2$ |
| D3PM uniform [18] (ours) | 1000 | $\leq 1.61 \pm 0.02$ | $3.6 \pm 0.4$ |
| D3PM NN ($L_{vb}$) (ours) | 1000 | $\leq 1.59 \pm 0.03$ | $3.1474 \pm 0.0002$ |
| D3PM absorbing ($L_{\lambda=0.01}$) (ours) | 1000 | $\leq 1.45 \pm 0.02$ | $3.4 \pm 0.3$ |
| D3PM uniform [18] (ours) | 256 | $\leq 1.68 \pm 0.01$ | $0.5801 \pm 0.0001$ |
| D3PM NN ($L_{vb}$) (ours) | 256 | $\leq 1.64 \pm 0.02$ | $0.813 \pm 0.002$ |
| D3PM absorbing ($L_{\lambda=0.01}$) (ours) | 256 | $\leq 1.47 \pm 0.03$ | $0.598 \pm 0.002$ |
| Transformer decoder (ours) | 256 | 1.37 | $0.3570 \pm 0.0002$ |
| Transformer decoder [1] | 256 | 1.18 | - |
| Transformer XL [9][†] | 256 | 1.08 | - |
| D3PM uniform [18] (ours) | 20 | $\leq 1.79 \pm 0.03$ | $0.0771 \pm 0.0005$ |
| D3PM NN ($L_{vb}$) (ours) | 20 | $\leq 1.75 \pm 0.02$ | $0.1110 \pm 0.0001$ |
| D3PM absorbing ($L_{\lambda=0.01}$) (ours) | 20 | $\leq 1.56 \pm 0.04$ | $0.0785 \pm 0.0003$ |

---

[6]https://github.com/google/sentencepiece

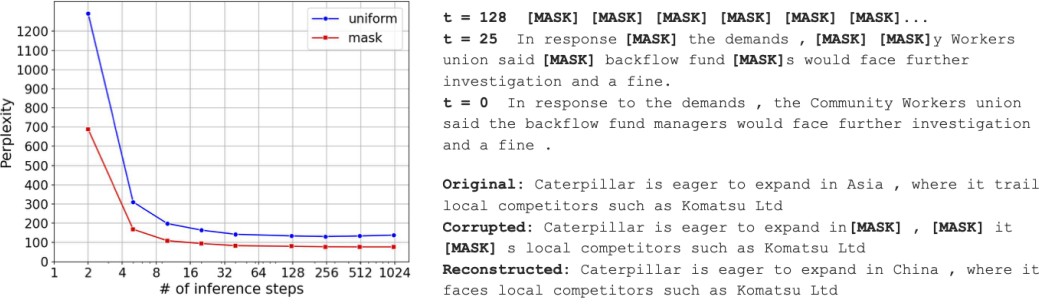

Figure 2: Left: perplexity v.s. sampling iterations for LM1B. Right: Using a trained D3PM absorbing model for LM1B to (top) generate new sentences and (bottom) reconstruct corrupted examples.

Table 2: Quantitative results on LM1B. Perplexity reported on the test set. Results are reported on two seeds. All models have context window length 128 and 12 layers unless otherwise noted. [†]Transformer XL is a 24 layer transformer. [‡]rounded for readability, see Appendix B.2.2.

| Metric: | Perplexity ($\downarrow$) | | | Sample time[‡] (s) ($\downarrow$) | | |
|---|---|---|---|---|---|---|
| inference steps: | 1000 | 128 | 64 | 1000 | 128 | 64 |
| D3PM uniform | $137.9 \pm 2.1$ | $139.2 \pm 1.2$ | $145.0 \pm 1.2$ | 1.82 | 0.21 | 0.08 |
| D3PM NN | $149.5 \pm 1.3$ | $158.6 \pm 2.2$ | $160.4 \pm 1.2$ | 21.29 | 6.69 | 5.88 |
| D3PM absorbing | $76.9 \pm 2.3$ | $80.1 \pm 1.2$ | $83.6 \pm 6.1$ | 1.90 | 0.19 | 0.10 |
| Transformer (ours) | - | 43.6 | - | - | 0.26 | - |
| Transformer XL [9][†] | - | 21.8 | - | - | - | - |

## 6 Image generation

We evaluate the performance of several D3PM models on the task of unconditional image generation with the dataset CIFAR-10 [25]. We follow Ho et al. [17] and use $T = 1000$ timesteps for all models and verify that for all models the forward process converges to the stationary distribution within $T$ steps, yielding a value of at most $L_T \approx 10^{-5}$ bits per dimension. We train three versions of D3PM with different transition matrices: doubly stochastic matrices with uniform transition probabilities (D3PM uniform) [18], transition matrices with an absorbing state located at R, G and B values of 128 (D3PM absorbing) and doubly stochastic discretized Gaussian transition matrices (D3PM Gauss). For the D3PM uniform model we experimented with a linear $\beta_t$ schedule as well as the cosine schedule as proposed in [18], with the cosine schedule producing the best results. For D3PM absorbing we use the schedule $\beta_t = (T - t + 1)^{-1}$ as also proposed in [36], which corresponds to increasing the probability of being in the absorbing state linearly over time. For D3PM Gauss we use the same linear schedule as in [17]. See Appendix B.1 for more details on the experimental setup.

Table 3 shows that for D3PM models trained with the $L_{\mathrm{vb}}$ objective, D3PM Gauss performs better than D3PM absorbing and uniform on all metrics: Inception score (IS), Frechet Inception Distance (FID) and negative log-likelihood (NLL). The IS score of the uniform and absorbing D3PM models are comparable, while the FID score and NLL of the D3PM absorbing model are slightly better. We trained both D3PM absorbing and D3PM Gauss with the alternative loss function $L_\lambda$ of (5), and we found $\lambda = 0.001$ to work best. We have also experimented with larger values of $\lambda$ and a model trained only with the auxiliary denoising term in (5). Although this led to a more rapid increase in performance early on in training, the NLL leveled off at higher values for larger $\lambda$ and the FID even started increasing again. The results show that the models trained with $L_\lambda$ perform significantly better than their counterparts trained with $L_{\mathrm{vb}}$. One explanation for this boost in performance is that the cross entropy term leads to gradient noise that varies less with the time step $t$, which is in contrast to the large change in magnitude of the $L_{t-1}$ terms in $L_{\mathrm{vb}}$ for smaller $t$, as demonstrated by Nichol and Dhariwal [28]. Finally, we achieve our best results by combining D3PM Gauss trained on $L_\lambda$ with a truncated logistic parameterization of the reverse process distribution $p_\theta(\widetilde{\boldsymbol{x}}_0|\boldsymbol{x}_t)$ (D3PM Gauss + logistic). Figure 3 shows samples from our best model (D3PM Gauss + logistic), as well as the D3PM absorbing model.

Table 3: Inception scores (IS), Frechet Inception Distance (FID) and negative log-likelihood (NLL) on the image dataset CIFAR-10. The NLL is reported on the test set in bits per dimension. We report our results as averages with standard deviations, obtained by training five models with different seeds.

| Model | IS ($\uparrow$) | FID ($\downarrow$) | NLL ($\downarrow$) |
|---|---|---|---|
| Sparse Transformer [8] | | | 2.80 |
| NCSN [38] | $8.87 \pm 0.12$ | 25.32 | |
| NCSNv2 [39] | $8.40 \pm 0.07$ | 10.87 | |
| StyleGAN2 + ADA [20] | $9.74 \pm 0.05$ | 3.26 | |
| Diffusion (original), $L_{\mathrm{vb}}$ [36] | | | $\leq 5.40$ |
| DDPM $L_{\mathrm{vb}}$ [17] | $7.67 \pm 0.13$ | 13.51 | $\leq 3.70$ |
| DDPM $L_{\mathrm{simple}}$ [17] | $9.46 \pm 0.11$ | 3.17 | $\leq 3.75$ |
| Improved DDPM $L_{\mathrm{vb}}$ [28] | | 11.47 | $\leq 2.94$ |
| Improved DDPM $L_{\mathrm{simple}}$ [28] | | 2.90 | $\leq 3.37$ |
| DDPM++ cont [40] | | 2.92 | 2.99 |
| NCSN++ cont. [40] | 9.89 | 2.20 | |
| D3PM uniform $L_{\mathrm{vb}}$ | $5.99 \pm 0.14$ | $51.27 \pm 2.15$ | $\leq 5.08 \pm 0.02$ |
| D3PM absorbing $L_{\mathrm{vb}}$ | $6.26 \pm 0.10$ | $41.28 \pm 0.65$ | $\leq 4.83 \pm 0.02$ |
| D3PM absorbing $L_{\lambda=0.001}$ | $6.78 \pm 0.08$ | $30.97 \pm 0.64$ | $\leq 4.40 \pm 0.02$ |
| D3PM Gauss $L_{\mathrm{vb}}$ | $7.75 \pm 0.13$ | $15.30 \pm 0.55$ | $\leq 3.966 \pm 0.005$ |
| D3PM Gauss $L_{\lambda=0.001}$ | $8.54 \pm 0.12$ | $8.34 \pm 0.10$ | $\leq 3.975 \pm 0.006$ |
| D3PM Gauss + logistic $L_{\lambda=0.001}$ | $8.56 \pm 0.10$ | $7.34 \pm 0.19$ | $\leq 3.435 \pm 0.007$ |

## 7 Related Work

Diffusion generative models were first proposed by Sohl-Dickstein et al. [36] and have gained renewed attention recently due to strong results on image and waveform generation [17, 7]. Recent works have proposed improvements for diffusion model training, including importance sampling of the ELBO, better noise schedules [28] and implicit diffusion models [37]. Several works have also drawn connections to score matching [45, 19, 38], leading to improved sampling algorithms in the continuous-time limit [40].

While most works have considered continuous diffusion models, discrete diffusion-like models were described in [36] and applied to text generation and image segmentation data in [18]. Some works [29, 27] have dealt with discrete data by embedding it in continuous space and leveraging Gaussian diffusion, but have not applied this to text. Seff et al. [35] considered generation of discrete structured objects using a diffusion-like Markov corruption process. Goyal et al. [15] proposed a diffusion-like model for images with a more flexible family of learned corruption processes. Ho et al. [17] also draws connections between diffusion and autoregressive models for continuous data.

For text, denoising autoencoders have a long history both in representation learning [2, 10] and more recently as generative models [47]. These closely resemble our absorbing state diffusion variants for

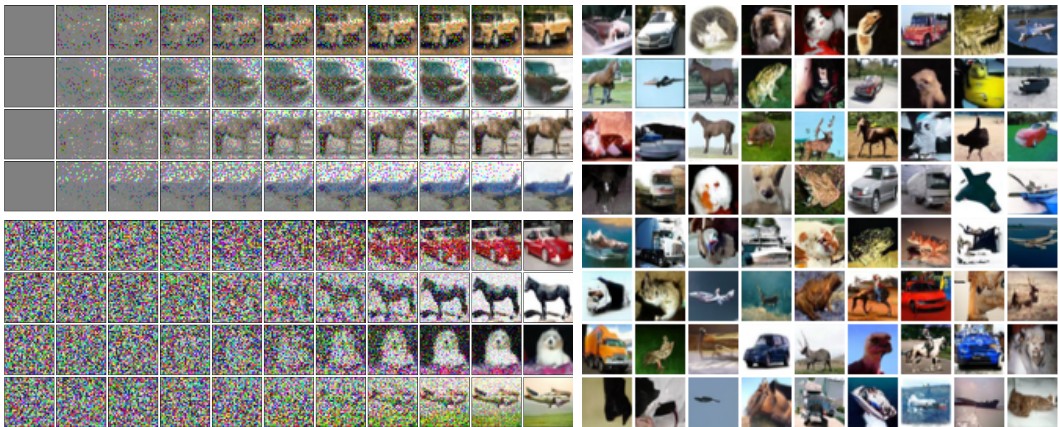

Figure 3: Left: progressive sampling at $t = 1000, 900, 800, ..., 0$ for D3PM absorbing (top) and D3PM Gauss + logistic (bottom), trained with $L_\lambda$ loss on CIFAR-10. These samples were cherry picked. Right: (non cherry picked) samples from the D3PM Gauss + logistic model.

a particular schedule and transition matrix (see Section 4), although our framing allows us to compute log-likelihoods and experiment with alternative transition matrices. Other works have considered non-autoregressive translation and speech transcription via insertion and deletion [16, 33], masking [13], and iteratively-refined sequence alignments [5, 34].

# 8    Discussion

We have presented D3PMs, a class of models that improves diffusion models for discrete data by defining new kinds of discrete corruption processes. We achieve strong empirical results relative to previous work on discrete diffusion models, even surpassing performance of continuous diffusion models in terms of log-likelihoods for image generation. While these results are promising, one limitation is that—like much other work on non-autoregressive generative models—our models are still inferior to strong autoregressive models like Transformer XL for text generation, and continuous diffusion models still yield stronger results on image quality. We expect that D3PMs can benefit further from the rapid development of continuous diffusion models [40, 28]. For example, further research in alternative losses for D3PM's can take inspiration from the reweighted $L_{\mathrm{simple}}$ objective used in [17], or the resampled variational bound in Nichol and Dhariwal [28]. Furthermore, D3PM's might benefit from increasing the number of timesteps and a more optimized noise schedule, as discussed in Nichol and Dhariwal [28]. Another limitation comes from the choice of evaluation metrics that we use (and that are standard for evaluation of generative models). Inception score and Frechet Inception Distance are based on neural networks that have been trained on a particular distribution of data, which is not representative for all use-cases, and focusing on average quality metrics may not accurately reflect performance across the wide diversity of settings where these generative models may be applied. This creates a risk of negative social impacts where advances disproportionately favor a subset of the population. Text generation models, including D3PMs, also present many challenges for responsible and reliable use. Prior works have highlighted the potential for misuse [24, 4], bias [46], and hallucination [48] in neural language models. D3PMs, like autoregressive language models, should be carefully evaluated along these axes before being deployed in a production setting. Going forward, we are excited about the space of possibilities that arise within the D3PM framework. We have found successes in leveraging the flexibility that comes from defining discrete corruption processes for discrete data, but we believe that there are many more possibilities that make use of richer forms of structure to define even more powerful discrete diffusion models.

## Acknowledgments and Disclosure of Funding

We would like to thank Hugo Larochelle for providing high-level feedback during the project, and Ben Poole for reviewing a draft version of this manuscript. We would also like to thank Julia Kreutzer and Xavier Garcia for helpful conversations about language experiments, and Daniel Watson for early discussions about discrete diffusion. We, the authors, declare to have no competing interests. The research conducted for this paper was entirely supported by Google.

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
