# A  Additional details regarding D3PMs

## A.1  Doubly-stochastic matrices

As discussed in Section 3.1, there are two constraints on $\boldsymbol{Q}_t$ that allow it to be used within a D3PM: the rows of $\boldsymbol{Q}_t$ must sum to one to conserve probability mass, and the rows of $\overline{\boldsymbol{Q}}_t = \boldsymbol{Q}_1 \boldsymbol{Q}_2 \dots \boldsymbol{Q}_t$ must converge to a known stationary distribution as $t$ becomes large. Technically, it is also possible to use a learned prior $p_\theta(\boldsymbol{x}_T)$, but assuming this is still modeled under a conditional independence assumption, $q(\boldsymbol{x}_T|\boldsymbol{x}_0)$ must still be close to a stationary distribution for the $L_T$ loss term to be small.

One way to ensure that this occurs is to chose $\boldsymbol{Q}_t$ as increasing powers of a doubly stochastic base matrix $\boldsymbol{Q}$ (rows and columns sum to 1) with strictly positive entries. This is enough to ensure that $\boldsymbol{Q}$ is is irreducible and aperiodic and that product $\overline{\boldsymbol{Q}}_t$ converges as $t \to \infty$ to a uniform distribution over all states. To show this, consider $\pi_i = 1/K$ for $i = 1, ..., K$, and $\sum_{i=1}^{K} \boldsymbol{Q}_{i,:} = \mathbf{1}$ and $\sum_{j=1}^{K} \boldsymbol{Q}_{:,j} = \mathbf{1}$, then $[\boldsymbol{Q}\boldsymbol{\pi}]_i = \sum_{j=1}^{K} \boldsymbol{Q}_{i,j} \pi_j = 1/K \sum_{j=1}^{K} \boldsymbol{Q}_{i,j} = 1/K = \pi_i$, thus the uniform distribution is an eigenvector of the transition matrix with eigenvalue 1. Convergence to this distribution follows from the Perron-Frobenius theorem for positive square matrices.

More generally, a similar argument shows that even for $\boldsymbol{Q}_t$ that are not powers of the same base matrix, as long as each $\boldsymbol{Q}_t$ is doubly stochastic, irreducible, and aperiodic, the uniform distribution is the only possible stationary distribution, and as long as the second largest eigenvalue of $\boldsymbol{Q}_t$ is bounded below, the cumulative product $\overline{\boldsymbol{Q}}_t$ will converge to the uniform distribution. In practice, we choose $\boldsymbol{Q}_t$ to add more noise as $t$ increases, which ensures that $\overline{\boldsymbol{Q}}_T$ is very close to reaching a uniform stationary distribution.

## A.2  More details on possible choices of Markov transition matrices

### A.2.1  Uniform diffusion

The transition matrix described by Sohl-Dickstein et al. [17] for the binary case, and extended by Hoogeboom et al. [9], to the categorical case, can be represented using the following $K \times K$ transition matrix

$$[\boldsymbol{Q}_t]_{ij} = \begin{cases} 1 - \frac{K-1}{K}\beta_t & \text{if} \quad i = j \\ \frac{1}{K}\beta_t & \text{if} \quad i \neq j \end{cases}, \tag{6}$$

This transition matrix can also be written as $(1 - \beta_t)I + \beta_t \mathbb{1}\mathbb{1}^T/K$, where $\mathbb{1}$ is a column vector of all ones.

### A.2.2  Diffusion with an absorbing state

For our diffusion models with an absorbing state $m$, we use the following matrix:

$$[\boldsymbol{Q}_t]_{ij} = \begin{cases} 1 & \text{if} \quad i = j = m \\ 1 - \beta_t & \text{if} \quad i = j \neq m \\ \beta_t & \text{if} \quad j = m, i \neq m \end{cases} \tag{7}$$

The transition matrix can also be written as $(1 - \beta_t)I + \beta_t \mathbb{1}e_m^T$, where $e_m$ is a vector with a one on the absorbing state $m$ and zeros elsewhere. Since $m$ is an absorbing state, the corruption process converges not to a uniform distribution but to the point-mass distribution on $m$.

For text generation, we let $m$ be the [MASK] token at index $K - 1$; this leads to a BERT-like training objective, which masks tokens according to some schedule and learns to denoise them iteratively (see Section 4). For image generation, we set $m$ to the gray RGB pixel $(128, 128, 128)$ at index $K//2$.

### A.2.3 Discretized Gaussian transition matrices

For our D3PM models applied to ordinal data, inspired by continuous-space diffusion models, we use the following $K \times K$ matrix:

$$[\boldsymbol{Q}_t]_{ij} = \begin{cases} \frac{\exp\left(-\frac{4|i-j|^2}{(K-1)^2\beta_t}\right)}{\sum_{n=-(K-1)}^{K-1} \exp\left(-\frac{4n^2}{(K-1)^2\beta_t}\right)} & \text{if} \quad i \neq j \\ 1 - \sum_{l=0, l\neq i}^{K-1}[\boldsymbol{Q}_t]_{il} & \text{if} \quad i = j \end{cases} \tag{8}$$

Normalization is ensured by assigning the diagonal values to one minus the sum of each row (not including the diagonal entry). Note that due to the normalization of the off-diagonal values over the range $\{-K+1, ..., K-1\}$ the sum of each row excluding the diagonal entry is always smaller than 1. The result yields an irreducible doubly stochastic matrix and a forward process with a uniform stationary distribution. Similar to the continuous Gaussian diffusion model, the parameters $\beta_t$ influence the variance of the forward process distributions.

### A.2.4 Structured diffusion in text: using word-embedding distance to introduce locality

For text, we construct a $k$-nearest neighbor adjacency matrix

$$[\mathbf{G}]_{ij} = 1 \text{ if } w_i \text{ is a k-nearest neighbor of } w_j \text{ else } 0$$

constructed from a pre-trained embedding space over the vocabulary. Then we consider a symmetrized adjacency matrix of the form $\mathbf{A} = (\mathbf{G} + \mathbf{G}^T)/(2k)$ where $k$ is the number of nearest neighbors of each node, and finally construct a doubly stochastic rate matrix with

$$[\boldsymbol{R}]_{ij} = \begin{cases} -\sum_{l\neq i} A_{il} & \text{if} \quad i = j \\ A_{ij} & \text{otherwise} \end{cases} \tag{9}$$

Our final transition matrix is constructed as a matrix exponential of this rate matrix:

$$\mathbf{Q}_t = \exp(\alpha_t \mathbf{R}) = \sum_{n=0}^{\infty} \frac{\alpha_t^n}{n!}\boldsymbol{R}^n$$

Since $\boldsymbol{R}$ is symmetric and sums to zero along each row, $\mathbf{Q}_t$ is doubly stochastic, which ensures we have a uniform stationary distribution (as long as $G$ is connected). Increasing $\alpha_t$ over time allows us to add more noise for larger values of $t$.

Assuming word embeddings are some metric for syntactic or semantic similarity, this results in a corruption process that gradually moves away from the ground-truth sentence, swapping words with nearest-neighbors in embedding space. For character level modeling, this is a graph over characters, which more often transitions for instance from vowels to other vowels than from vowels to consonants. For words, this could transition between semantically similar words.

For example, in Figure 4, we construct the forward process to diffuse from "dog" to "cat" or "cow", which are nearby in embedding space, but not to more distant words. We can either bootstrap this process by updating the transition matrix $\boldsymbol{Q}$ dynamically during training, or use pretrained embeddings; we use pretrained embeddings for all of our experiments. Specifically, we train an autoregressive language model on the dataset in question (either text8 or LM1B) with randomly initialized word embeddings (768 dimensional in most cases), and then use $L^2$ or cosine similarity to compute the $k$-nearest neighbors of each token. We transition preferentially to these tokens, although the matrix exponential in theory allows transitions to any other token. We choose $k$ large enough so the resulting graph is connected.

### A.2.5 Band-diagonal transitions

A class of transition matrices that introduce local, ordinal inductive biases for structured data are band-diagonal transition matrices which only allow the corruption process to transition locally between states and biases the reverse process towards local iterative refinement. For example, in images, this can be used to allow transitions only between adjacent pixel values.

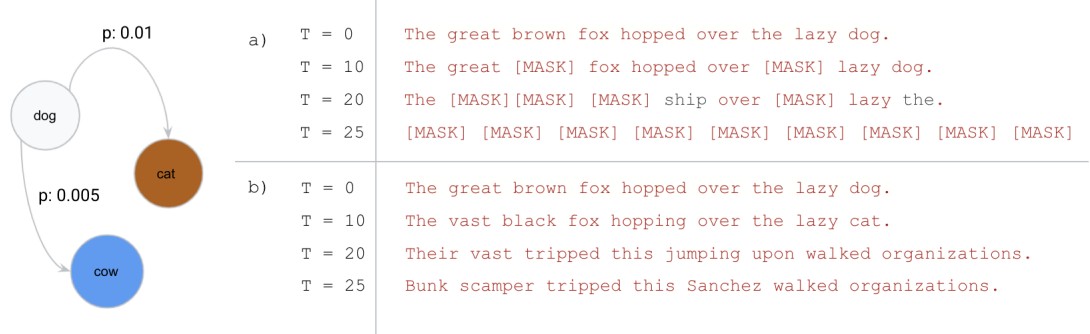

Figure 4: Two examples of noise schedules transforming text data. The top is a BERT-like absorbing + uniform diffusion which replaces tokens with [MASK] tokens (and occasionally with any other token, in black). The bottom is nearest-neighbor diffusion in embedding space. At left represents a possible column in the transition matrix.

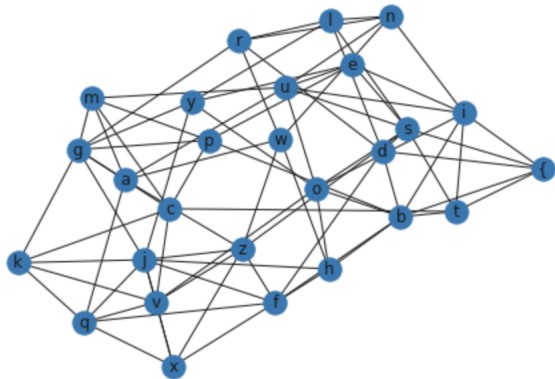

Figure 5: The character-level symmetrized 5-NN graph.

$$[\boldsymbol{Q}_t]_{ij} = \begin{cases} \frac{1}{K}\beta_t & \text{if} \quad 0 < |i-j| \le v \\ 1 - \sum_{l \ne i} Q_{il} & \text{if} \quad i = j \end{cases} \tag{10}$$

where $v$ is the number of nonzero off-diagonal elements of $\boldsymbol{Q}$ above (and below) the main diagonal. Note that this is a doubly stochastic matrix, so the stationary distribution is uniform. We do not use these in our experiments.

### A.2.6 Combinations of absorbing diffusion and other diffusion

A few ablations in Appendix B.2.1 consider transition matrices that combine absorbing-state or nearest-neighbor and uniform D3PM models. For instance, an absorbing-uniform transition matrix can be constructed $\boldsymbol{Q} = \alpha \mathbb{1}e_m^T + \beta \mathbb{1}\mathbb{1}^T/K + (1 - \alpha - \beta)I$, where $e_m$ is a one-hot vector on the [MASK] token.

### A.3 Generative Masked Language Models are Diffusion Models

Generative Masked Language Models [5, 21] are generative models that generate text from a sequence of [MASK] tokens. These are usually trained by sampling a sequence $\boldsymbol{x}_0$, masking tokens according to some schedule, and learning to predict the masked tokens given context. The actual masking procedure can either be done independently, i.e. by masking each token with probability $p = k/T$,

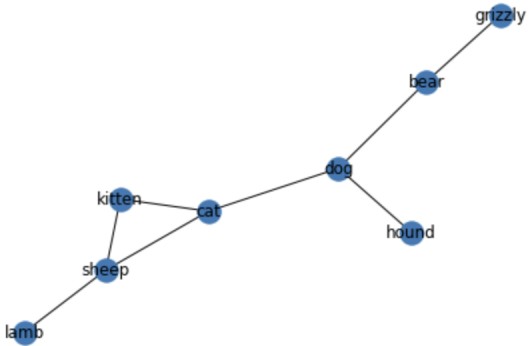

Figure 6: Subgraph of a word-level NN graph.

like Devlin et al. [3], or by sampling exactly $k$ tokens. The usual objective is[7]:

$$\min -\mathbb{E}_{q(\boldsymbol{x}_0)} \left[ \mathbb{E}_{k \in [1...|\boldsymbol{x}_0|]} \left[ \frac{1}{k} \mathbb{E}_{\boldsymbol{x}_k \text{ with } k \text{ masked tokens}} \left[ \sum_{i \text{ with } [\boldsymbol{x}_k]_i = m} \log p_\theta([\boldsymbol{x}_0]_i | \boldsymbol{x}_k) \right] \right] \right] \quad (11)$$

where we first sample a datapoint $\boldsymbol{x}_0$, sample a number of tokens to mask $k$ (either uniformly or according to some schedule), then mask that many tokens at random and compute a cross entropy loss over those masked tokens. We claim that this training objective is a (reweighted) absorbing-state D3PM objective with a particular noise schedule and the $\boldsymbol{x}_0$-parameterization from 3.3 (and indeed, that any absorbing-state D3PM model with [MASK] as the absorbing state will be a reweighted version of this loss with different weights assigned to different numbers of masked tokens $k$).

Consider a D3PM with a schedule that masks tokens with probability $\beta_t$. The reverse process predicts $\widetilde{p}_\theta(\widetilde{\boldsymbol{x}_0}|\boldsymbol{x}_t)$, then uses the forward process to compute $p_\theta(\boldsymbol{x}_{t-1}|\boldsymbol{x}_t) \propto \sum q(\boldsymbol{x}_{t-1}, \boldsymbol{x}_t|\widetilde{\boldsymbol{x}_0})\widetilde{p}_\theta(\widetilde{\boldsymbol{x}_0}|\boldsymbol{x}_t)$. In the particular case of absorbing-state diffusion, for each masked token $[\boldsymbol{x}_t]_i = m$ in $\boldsymbol{x}_t$, we thus have

$$p_\theta([\boldsymbol{x}_{t-1}]_i | \boldsymbol{x}_t) \propto \begin{cases} [\beta_t \prod_{s<t}(1-\beta_s)]\widetilde{p}_\theta([\widetilde{\boldsymbol{x}}_0]_i = [\boldsymbol{x}_0]_i | \boldsymbol{x}_t) & \text{for } [\boldsymbol{x}_{t-1}]_i = [\boldsymbol{x}_0]_i \neq m \\ 1 - \prod_{s\leq t}(1-\beta_s) & \text{for } [\boldsymbol{x}_{t-1}]_i = m \end{cases}$$

We note that for each unmasked token $[\boldsymbol{x}_t]_i = [\boldsymbol{x}_0]_i$, the KL-divergence is zero since unmasked tokens cannot make any other type of transition other than becoming masked. Also, the term in the KL divergence due to the probability of mask transitions is a constant, since mask transitions are independent of the model parameters $\theta$. Our $L_t$ term is then

$$D_{\text{KL}}[q(\boldsymbol{x}_{t-1}|\boldsymbol{x}_t, \boldsymbol{x}_0)||p_\theta(\boldsymbol{x}_{t-1}|\boldsymbol{x}_t)] = -\left[\beta_t \prod_{s<t}(1-\beta_s)\right] \sum_{i \text{ with } [\boldsymbol{x}_t]_i=m} \log \widetilde{p}_\theta([\boldsymbol{x}_0]_i | \boldsymbol{x}_t) + C$$

where $C$ is independent of $\theta$ and the sum is taken over the masked tokens in $\boldsymbol{x}_t$. For example, if we use $\beta(t) = 1/(T - t + 1)$ from Sohl-Dickstein et al. [17], $\beta_t \prod_{i=0}^{t-1}(1 - \beta_i) = 1/T$ and $1 - \prod_{i=0}^{t}(1 - \beta_i) = (t - 1)/T$, so $q([\boldsymbol{x}_{t-1}]_i = [\boldsymbol{x}_0]_i | [\boldsymbol{x}_t]_i = m, \boldsymbol{x}_0) = 1/t$ for non-mask tokens and we can simplify our $L_t$ objective to

$$D_{\text{KL}}[q(\boldsymbol{x}_{t-1}|\boldsymbol{x}_t, \boldsymbol{x}_0)||p_\theta(\boldsymbol{x}_{t-1}|\boldsymbol{x}_t)] = -\left[\frac{1}{t} \sum_{i \text{ with } [\boldsymbol{x}_t]_i=m} \log \widetilde{p}_\theta([\boldsymbol{x}_0]_i | \boldsymbol{x}_t)\right] + C$$

where $\boldsymbol{x}_t$ masks tokens independently and uniformly with probability $t/T$. The $L_T$ term in our ELBO is 0 for the $1/(T - t + 1)$ schedule, so the full objective (up to a constant) reduces to

---

[7]Sometimes the loss is un-normalized or normalized by the full sequence length.

$$\mathbb{E}_{q(\boldsymbol{x}_0)}\left[-\sum_{t=2}^{T}\frac{1}{t}\mathbb{E}_{q(\boldsymbol{x}_t|\boldsymbol{x}_0)}\left[\sum_{i \text{ with } [\boldsymbol{x}_t]_i=m}\log p_\theta([\boldsymbol{x}_0]_i|\boldsymbol{x}_t)]\right]\right.$$

$$\left.-\mathbb{E}_{q(\boldsymbol{x}_1|\boldsymbol{x}_0)}\left[\sum_{i \text{ with } [\boldsymbol{x}_1]_i=m}\log p_\theta([\boldsymbol{x}_0]_i|\boldsymbol{x}_1)]\right]\right]$$

$$= -\mathbb{E}_{q(\boldsymbol{x}_0)}\left[\sum_{t=1}^{T}\frac{1}{t}\mathbb{E}_{q(\boldsymbol{x}_t|\boldsymbol{x}_0)}\left[\sum_{i \text{ with } [\boldsymbol{x}_t]_i=m}\log p_\theta([\boldsymbol{x}_0]_i|\boldsymbol{x}_t)]\right]\right] \tag{12}$$

Note that while this looks very similar to Equation 11 (with each term reweighted by $1/t$, the expected number of masked tokens) it is not exactly identical since masking is computed independently per-token position (instead of choosing exactly $k$ tokens to mask). This is an entirely practical way to do masking (and indeed some methods implement it this way).

Furthermore, since the masking probability varies linearly as $1-\prod(1-\beta_t) = t/T$, this is very close to uniformly sampling the number of masked tokens $k$, but $k$ is actually drawn from a mixture of binomial distributions, i.e.

$$= -\mathbb{E}_{q(\boldsymbol{x}_0)}\left[\mathbb{E}_{k\in[1...|X|]}\left[\mathbb{E}_{\boldsymbol{x}_k \text{ with } k \text{ masked tokens}}\left[\alpha(k)\sum_{i \text{ with } [\boldsymbol{x}_k]_i=m}\log p_\theta([\boldsymbol{x}_0]_i|\boldsymbol{x}_k)]\right]\right]\right] \tag{13}$$

$$\alpha(k) = q(\boldsymbol{x}_t \text{ has } k \text{ masked tokens}|\boldsymbol{x}_0 \text{ has } n \text{ tokens}) = \frac{1}{T}\sum_{t=1}^{T}\binom{n}{k}\left(\frac{t}{T}\right)^{n-1}\left(1-\frac{t}{T}\right)^{n-k} \tag{14}$$

which is very close to uniform weight over terms, but slightly downweights terms near 0 and $T$. By upweighting terms near the boundary, you could in theory make this exactly uniform and thus exactly recover Equation 11. For instance, for 50 categories, absorbing-state diffusion produces the weighting shown in Figure 7.

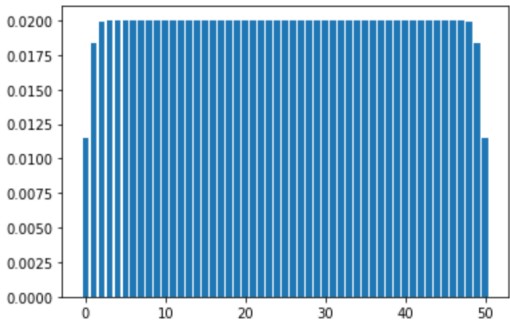

Figure 7: Plot of the probabilities of having $k$ tokens masked out of a length-50 sequence under a D3PM absorbing schedule with $T = 50$ steps, which is very similar to the uniform weighting used by Ghazvininejad et al. [5].

## A.4  Scaling to a large number of categories

When the number of categories $K$ is large, it can quickly become impractical to store all of the transition matrices $\boldsymbol{Q}_t$ in memory, as the memory usage grows like $O(K^2 T)$. And even if there is an algorithm to compute individual step matrices $\boldsymbol{Q}_t$ on demand, it may or may not be possible to do the same for the cumulative products $\overline{\boldsymbol{Q}}_t$. We propose two approaches to scaling D3PMs to large numbers of categories that ensure cumulative products are efficient: using low-rank corruption and using matrix exponentials.

### A.4.1 Low-rank corruption

In the low-rank case, we consider structuring our transition matrices as

$$\boldsymbol{Q}_t = \beta_t \boldsymbol{A}_t + (1 - \beta_t)\boldsymbol{I}, \tag{15}$$

where each $\boldsymbol{A}_t$ is a diagonalizable low-rank matrix with the same nonzero eigenvectors. In particular, recall that both absorbing-state diffusion and uniform diffusion have this form: for uniform diffusion, $\boldsymbol{A}_t^{\text{uniform}} = \mathbb{1}\mathbb{1}^T/K$, and for absorbing-state diffusion $\boldsymbol{A}_t^{\text{abs}} = \mathbb{1}e_m^T$ where $e_m$ is a one-hot vector on the absorbing state. Since products of $\boldsymbol{A}_t$'s are also low rank, the cumulative products $\overline{\boldsymbol{Q}}_t$ can be efficiently precomputed and stored using a much smaller amount of memory $O(r^2 T)$ where $r = \text{rank}(\boldsymbol{A}_t)$.

As an illustrative example, we describe in more detail how to efficiently represent uniform and absorbing-state transition matrices using the low-rank structure.

To compute products of uniform transition matrices (i.e. $\prod_i (1 - \beta_i)I + \beta_i \mathbb{1}\mathbb{1}^T/K$), we can take advantage of the useful fact that products of matrices of the form $\alpha I + \beta \mathbb{1}\mathbb{1}^T$ also have this same form: $I^2 = I$ and $\left(\beta\mathbb{1}\mathbb{1}^T\right)^2 = \beta^2 K \mathbb{1}\mathbb{1}^T$. We can thus treat this as a formal polynomial in one variable $X = (\mathbb{1}\mathbb{1}^T/K)$. Then products can be computed as $\prod_i \left[(1 - \beta_i) + \beta_i X\right]$ over the quotient ring $\mathbb{R}[X]/(X^2 - X)$, since $X^2 = X$. Functionally, this means you can instantiate a polynomial $(1 - \beta_i) + \beta_i X$ and repeatedly perform ordinary polynomial multiplication over $\mathbb{R}[X]$ for the $t < T$ timesteps. After each multiplication, the higher-order terms are reduced by $X^2 = X$, leaving a polynomial of degree 1 where the $X$ term has coefficient given by the sum of all higher-order terms. This can be computed with the convenient *np.polynomial* module.

Similarly, the transition matrices for D3PM absorbing can be computed in closed form. Fundamentally, in each step, we transition to a [MASK] token with probability $\beta_t$ and stay the same with probability $1 - \beta_t$. Since the [MASK] state is absorbing, after $t$ steps, the only operative quantity is the probability of not yet having transitioned to the [MASK] state, given by $\widetilde{\alpha}_t = \prod_{i=0}^{t}(1 - \beta_i)$. Hence for D3PM absorbing, $\overline{\boldsymbol{Q}} = \tilde{\alpha}_t I + (1 - \widetilde{\alpha}_t)\mathbb{1}e_m^T$ where $e_m$ is a one-hot vector on the [MASK] token.

### A.4.2 Matrix exponentials

In the matrix exponential case, we specify our transition matrices as

$$\boldsymbol{Q}_t = \exp(\alpha_t \boldsymbol{R}) = \sum_{n=0}^{\infty} \frac{\alpha_t^n}{n!} \boldsymbol{R}^n, \qquad \overline{\boldsymbol{Q}}_t = \exp\left(\left(\textstyle\sum_{s \le t} \alpha_s\right)\boldsymbol{R}\right), \tag{16}$$

where $\boldsymbol{R}$ is a *transition rate matrix* and $\exp$ denotes the matrix exponential operation; the similar form for $\boldsymbol{Q}_t$ and $\overline{\boldsymbol{Q}}_t$ is a consequence of the "exponential of sums" property for commuting matrices. For efficiency, we further assume that each of the $\alpha_t$ is an integer multiple $n_t \alpha_\star$ of some common factor $\alpha_\star$, and precompute matrices $\exp(2^k \alpha_\star \boldsymbol{R})$ for $0 \le k \le \log_2(\overline{\alpha}_T/\alpha_\star)$, where $\overline{\alpha}_T = \sum_{t<T} \alpha_t$, taking space $O(K^2 \log(\overline{\alpha}_T/\alpha_\star))$. Then, to compute matrix-vector products with $\boldsymbol{Q}_t$ or $\overline{\boldsymbol{Q}}_t$, we can iteratively take products with a subset of these precomputed matrices based on the digits of a binary expansion of the desired multiple $n_t$ in time $O(K^2 \log(\overline{\alpha}_T/\alpha_\star))$.[8]

As long as $\boldsymbol{R}$ has non-positive off-diagonal entries and sums to zero along each row, the matrix exponential produces a valid transition matrix $\boldsymbol{Q}_t$; convergence to a specific stationary distribution can also be ensured by controlling the eigenvectors. In particular, if every column also sums to zero, the resulting $\boldsymbol{Q}_t$ will be doubly stochastic and will thus have a uniform stationary distribution.

We note that this parameterization can be viewed as a discretization of a continuous-time discrete-space Markov processes; we describe this connection in more detail in the following section.

### A.5 Continuous-time Markov process transition rates

Following Feller [4], we define a continuous-time discrete-space Markov process as a collection of random variables $\{\boldsymbol{x}_t\}_{t>0}$ parameterized by $t \in \mathbb{R}^+$ and characterized by a Markov property

---

[8]This is closely related to the well-known "exponentiation-by-squaring" technique.

$(\boldsymbol{x}_t \perp \boldsymbol{x}_s \mid \boldsymbol{x}_\tau$ if $t < \tau < s)$, a transition probability matrix $\Pi(t) \in \mathbb{R}^{N \times N}$ where $N$ is the cardinality of $\boldsymbol{x}_t$, and a set of transition rates $\boldsymbol{\gamma}_i(t)$.

A conceptual way to understand these processes is to imagine a continuous Poisson process occurring in each state $i$ at rate $\boldsymbol{\gamma}_i(t)$ determining when a transition between states occurs. When a transition occurs (at time $t$), a Markov transition occurs between states $i$ and $j$ with probability $\Pi_{ij}(t)$. Many common stochastic processes fall into this family, including Poisson processes. Like in the case of stochastic differential equations (Song et al. [18]), we can derive a set of Kolomogorov equations (or Fokker-Planck equations in the continuous-state space case) that determine the marginal probability $\partial q_{ij}(\tau, t)$ of ending up in state $j$ at time $t$ having started in state $i$ at time $s$. The general form of the Kolmogorov forward equations is

$$\frac{\partial q_{ij}(\tau, t)}{\partial t} = -\boldsymbol{\gamma}_k(t) q_i(\tau, t) + \sum_j \boldsymbol{\gamma}_j(t) \Pi_{kj}(t) q_{ik}(t)$$

Now we can state and prove a theorem connecting continuous time Markov processes and matrix exponentials.

**Theorem 1.** *Let $\{\boldsymbol{x}_t\}_{t \geq 0}$ be a discrete-space, continuous-time Markov process with (possibly time-dependent) transition probability matrix $\Pi(t)$ and transition rates $\boldsymbol{\gamma}_i(t)$. Then for a particle with an initial distribution $q(\boldsymbol{x}_s)$ at time $s$, the probability of ending in state $j$ at time $t$ is*

$$q(\boldsymbol{x}_t | \boldsymbol{x}_s) = \exp\left(\int_s^t \mathrm{diag}(\boldsymbol{\gamma}(\tau))(\Pi(\tau) - I)\, d\tau\right) q(\boldsymbol{x}_s)$$

*where $\exp$ is the matrix exponential and we view $q(\boldsymbol{x}_t)$ and $\boldsymbol{\gamma}(t)$ as vectors in $\mathbb{R}^N$.*

*Proof (sketch).* From the Kolmogorov equations for continuous-time Markov processes, we have the ODE

$$\frac{\partial q(\boldsymbol{x}_t | \boldsymbol{x}_s)}{\partial t} = \mathrm{diag}(\boldsymbol{\gamma}(t))(\Pi(t) - I) q(\boldsymbol{x}_t | \boldsymbol{x}_s)$$

where $\Pi(t)$ is the transition probability matrix. Solving this as a first-order ODE using integrating factors yields the desired equation. $\qquad \square$

We note that, if $\Pi(t) = \Pi$ is independent of $t$ and $\boldsymbol{\gamma}(s) = \gamma(s)\mathbf{r}$ for some scalar function $\gamma : \mathbb{R} \to \mathbb{R}$ and vector $\mathbf{r} \in \mathbb{R}^N$, this simplifies to exactly our matrix exponential parameterization with

$$\mathbf{R} = \mathrm{diag}(\mathbf{r})(\Pi - I).$$

where we set

$$\alpha_t = \int_{t-1}^t \gamma(t)\, dt.$$

In other words, the $\alpha_t$ parameters in Equation 16 correspond to a discretization of the cumulative transition rate of a continuous-time process.

### A.6 Continuous-limit of schedule from Sohl-Dickstein et al. [17]

Consider for example the schedule described by Sohl-Dickstein et al. [17] for Bernoulli variables $\beta_t = 1/(T - t + 1)$, i.e. the Bernoulli variable would stay the same with probability $1 - \beta_t = (T - t)/(T - t + 1)$ and transition with probability $\beta_t$. In this section, we show that a D3PM absorbing or D3PM uniform process with this schedule is exactly a discretization of a continuous-time jump process of the form described in Theorem 1.

We start by observing that both absorbing-state and uniform D3PM transition matrices can be expressed equivalently as matrix exponentials. In the uniform case, we have

$$Q_t = \exp(\alpha_t \mathbf{R}_{\mathrm{unif}}) = \exp\left(\alpha_t\left(\frac{1}{K}\mathbb{1}\mathbb{1}^T - I\right)\right) = \exp(-\alpha_t)I + (1 - \exp(-\alpha_t))\frac{1}{K}\mathbb{1}\mathbb{1}^T,$$

and in the absorbing case we have

$$Q_t = \exp(\alpha_t \mathbf{R}_{\text{abs}}) = \exp\left(\alpha_t\left(\mathbb{1}\mathbf{e}_m^T - I\right)\right) = \exp(-\alpha_t)I + (1 - \exp(-\alpha_t))\mathbb{1}\mathbf{e}_m^T.$$

In either case, by setting this equal to the explicit forms in Appendix A.2, we obtain the relationship

$$\beta_t = 1 - \exp(-\alpha_t)$$

where $\beta_t$ is defined as in Appendix A.2, and $\alpha_t$ is the matrix exponential coefficient as used in the previous section. Using the correspondence discussed in the previous section, we also know

$$\alpha_t = \int_{t-1}^{t} \gamma(s)\,ds$$

for the continuous-time transition rate function $\gamma(s)$. Defining $\beta_t = 1/(T - t + 1)$, we have

$$1 - \beta_t = 1 - \frac{1}{(T - t + 1)} = \frac{T - t}{T - t + 1} = \exp\left(-\int_{t-1}^{t} \gamma(\tau)d\tau\right)$$

Denoting the anti-derivative $\int \gamma(t) = F(t)$, we have $\log(T-t) - \log(T-t+1) = -F(t) + F(t-1)$, so we can deduce $F(t) = -\log(T - t)$ (up to a constant offset). Taking a derivative then yields $\gamma(t) = 1/(T - t)$, which has the same form as the original schedule but is now interpreted as a continuously-varying rate function instead of a probability (and is also shifted by 1 unit in time). Intuitively, we can interpret this as a schedule which assigns uniform probability of a transition occurring over the remaining time, but instead of dividing it between $T - t + 1$ discrete steps, we divide it across a continuous interval of size $T - t$. We note that using larger values of $T$ is equivalent to performing a finer discretization on a scaled version of this continuous-time process.

## A.7 Mutual-information-based noise schedule

An important part of designing the forward process for a diffusion process is to specify the *noise schedule*: how much noise is added at each step $t$ such that after $T$ steps the process has (approximately) reached the stationary distribution of the transition matrix. Previous work on continuous-state diffusion models [8, 11, 18] has focused on controlling the variance of the continuous noise added at each step, but in a discrete state space it is less obvious how to measure or control the level of noise added.

For uniform or absorbing-state transition matrices, once a single transition occurs, all information about the original data point is lost. In this case, the schedule introduced by Sohl-Dickstein et al. [17] is a natural choice, since it is designed to make this first transition for $t/T$ of the elements by time $t$. However, when the transition matrix imposes additional structure on the transitions, such as for our token-embedding based transition matrix, it is not sufficient to perturb $t/T$ of the elements by time $t$, since the value at time $t$ may be highly correlated with the value at time $t - 1$ even after a transition occurs; we thus explore using mutual information to quantify how much noise has been added. Here we describe the mutual-information-based schedules in more detail. We focus on transition matrices that are parameterized as matrix exponentials, i.e. they have the form

$$\boldsymbol{Q}_t = \exp(\alpha_t \boldsymbol{R}) = \sum_{n=0}^{\infty} \frac{\alpha_t^n}{n!} \boldsymbol{R}^n, \qquad \overline{\boldsymbol{Q}}_t = \exp\left(\left(\sum_{s \le t} \alpha_s\right)\boldsymbol{R}\right) = \exp\left(\bar{\alpha}_t \boldsymbol{R}\right).$$

Inspired by the schedule introduced by Sohl-Dickstein et al. [17], we consider setting our $\alpha_t$ such that $\frac{t}{T}$ of the information about $p(\boldsymbol{x}_0)$ has been lost by time $t$. Our goal is to find exponents such that

$$\frac{t}{T} = 1 - \frac{I(\boldsymbol{x}_t; \boldsymbol{x}_0)}{H(\boldsymbol{x}_0)} = \frac{H(\boldsymbol{x}_0, \boldsymbol{x}_t) - H(\boldsymbol{x}_t)}{H(\boldsymbol{x}_0)} = \frac{\sum_{\boldsymbol{x}_0, \boldsymbol{x}_t} p(\boldsymbol{x}_0)q(\boldsymbol{x}_t|\boldsymbol{x}_0)\log \frac{q(\boldsymbol{x}_t|\boldsymbol{x}_0)}{\sum_{\boldsymbol{x}_0'} p(\boldsymbol{x}_0')q(\boldsymbol{x}_t|\boldsymbol{x}_0')}}{\sum_{\boldsymbol{x}_0} p(\boldsymbol{x}_0)\log p(\boldsymbol{x}_0)} \quad (17)$$

where $H$ denotes the entropy of a random variable, and $p(\boldsymbol{x}_0)$ denotes the distribution of a randomly chosen token in the data.

In practice, we estimate $p(\boldsymbol{x}_0)$ by computing empirical frequencies over the training set, and compute the value of the right-hand side of 17 for transition matrices $\exp(\bar{\alpha}\boldsymbol{R})$ with 256 geometrically-spaced

exponents $\bar{\alpha}$ distributed in a large range (linear on a log scale between 1e-4 and 1e5). We then interpolate using a monotonic cubic spline to find the particular exponents $\bar{\alpha}_t$ that ensure the above property holds approximately, and round them so that they are all multiples of a common factor $\alpha_\star$ to ensure efficiency (as described in Appendix A.4). Finally, we set $\boldsymbol{Q}_t = \exp((\bar{\alpha}_t - \bar{\alpha}_{t-1})\boldsymbol{R})$.

It turns out that, for the specific case of absorbing-state diffusion with a [MASK] token, the mutual information schedule reduces to exactly the $(T-t+1)^{-1}$ schedule proposed by Sohl-Dickstein et al. [17]. To see this, let $m_t$ be the probability that a given value from time 0 has been replaced with [MASK] at time $t$. We note then that

$$H(\boldsymbol{x}_t) = \sum_{\boldsymbol{x}_0}(1-m_t)p(\boldsymbol{x}_0)\log\left((1-m_t)p(\boldsymbol{x}_0)\right) + m_t\log m_t$$

$$= (1-m_t)\sum_{\boldsymbol{x}_0}p(\boldsymbol{x}_0)\log p(\boldsymbol{x}_0) + (1-m_t)\log(1-m_t) + m_t\log m_t$$

where we have used the fact that a mask token has zero probability under the data distribution. We also have the joint entropy

$$H(\boldsymbol{x}_0, \boldsymbol{x}_t) = \sum_{\boldsymbol{x}_0}p(\boldsymbol{x}_0)\log p(\boldsymbol{x}_0) + m_t\log m_t + (1-m_t)\log(1-m_t).$$

We can then calculate

$$1 - \frac{I(\boldsymbol{x}_t;\boldsymbol{x}_0)}{H(\boldsymbol{x}_0)} = \frac{H(\boldsymbol{x}_0,\boldsymbol{x}_t) - H(\boldsymbol{x}_t)}{H(\boldsymbol{x}_0)}$$

$$= \frac{\sum_{\boldsymbol{x}_0}p(\boldsymbol{x}_0)\log p(\boldsymbol{x}_0) + m_t\log m_t + (1-m_t)\log(1-m_t)}{\sum_{\boldsymbol{x}_0}p(\boldsymbol{x}_0)\log p(\boldsymbol{x}_0)}$$

$$- \frac{(1-m)\sum_{\boldsymbol{x}_0}p(\boldsymbol{x}_0)\log p(\boldsymbol{x}_0) + (1-m_t)\log(1-m_t) + m_t\log m_t}{\sum_{\boldsymbol{x}_0}p(\boldsymbol{x}_0)\log p(\boldsymbol{x}_0)}$$

$$= \frac{m_t\sum_{\boldsymbol{x}_0}p(\boldsymbol{x}_0)\log p(\boldsymbol{x}_0)}{\sum_{\boldsymbol{x}_0}p(\boldsymbol{x}_0)\log p(\boldsymbol{x}_0)} = m_t.$$

It follows that the mutual information schedule for masks is one that ensures $m_t = q(\boldsymbol{x}_t = [\text{MASK}]|\boldsymbol{x}_0) = \frac{t}{T}$. But this is exactly the $(T-t+1)^{-1}$ schedule. To see this, let $\beta_t$ be the probability that a non-mask token becomes a mask token at time $t$, and note that $m_t = 1 - \prod_{s=1}^{t}(1-\beta_s)$. Thus,

$$\beta_t = 1 - \frac{1-m_t}{1-m_{t-1}} = 1 - \frac{1-\frac{t}{T}}{1-\frac{t-1}{T}} = 1 - \frac{T-t}{T-t+1} = \frac{(T-t+1)-(T-t)}{T-t+1} = \frac{1}{T-t+1}$$

as desired.

Interestingly, although the $(T-t+1)^{-1}$ schedule was designed for the case of a uniform transition matrix (an used for this purpose by Sohl-Dickstein et al. [17] and Hoogeboom et al. [9]), the $(T-t+1)^{-1}$ schedule is NOT in general identical to the mutual information schedule in that setting. We leave further investigation of these schedules to future work.

### A.8  Parameterizing the reverse process with a discretized truncated logistic distribution

For ordinal data such as images, we can instill an ordinal inductive bias in the logits of $\widetilde{p}_\theta(\widetilde{\boldsymbol{x}}_0|\boldsymbol{x}_t)$ by modeling them using a discretization of a distribution on real-valued numbers. In this paper we choose the underlying continuous distribution to be a truncated logistic distribution. The code below shows how we compute the logits for $\widetilde{p}_\theta(\widetilde{\boldsymbol{x}}_0|\boldsymbol{x}_t)$, given a location/mean and a log scale that were predicted by a neural network $\text{nn}_\theta$.

```
import jax.numpy as jnp

def get_logits_from_logistic_pars(loc, log_scale, num_classes):
  """Computes logits for an underlying logistic distribution."""

  # The loc and log_scale are assumed to be modeled for data re-scaled
```

```
8     # such that the values {0, ...,K-1} map to the interval [-1, 1].
9     # Shape of loc and log_scale: (batch_size, height, width, channels)
10    loc = jnp.expand_dims(loc, axis=-1)
11    log_scale = jnp.expand_dims(log_scale, axis=-1)
12
13    # Shift log_scale such that if it's zero the output distribution
14    # has a reasonable variance.
15    inv_scale = jnp.exp(- (log_scale - 2.))
16
17    bin_width = 2. / (num_classes - 1.)
18    bin_centers = jnp.linspace(start=-1., stop=1., num=num_classes,
19                               endpoint=True)
20    bin_centers = jnp.expand_dims(bin_centers,
21                                  axis=tuple(range(0, loc.ndim-1)))
22
23    bin_centers = bin_centers - loc
24    # Note that the edge bins corresponding to the values 0 and K-1
25    # don't get assigned all of the mass in the tails to +/- infinity.
26    # So the logits correspond to unnormalized log probabilites of a
27    # discretized truncated logistic distribution.
28    log_cdf_min = jax.nn.log_sigmoid(
29        inv_scale * (bin_centers - 0.5 * bin_width))
30    log_cdf_plus = jax.nn.log_sigmoid(
31        inv_scale * (bin_centers + 0.5 * bin_width))
32
33    logits = log_minus_exp(log_cdf_plus, log_cdf_min)
34
35    return logits
36
37
38 def log_minus_exp(a, b, epsilon=1.e-6):
39   """Computes the log(exp(a) - exp(b)) (b<a) in a numerically stable way."""
40
41   return a + jnp.log1p(-jnp.exp(b - a) + epsilon)
```

## A.9    Auxiliary loss

Here we show that, for some choices of forward process $q$, there are parameterizations $\widetilde{p}_\theta(x_0|x_t)$ that are optimal under any reweighting of the ELBO but not optimal under the auxiliary loss. This occurs because the ELBO only supervises $\widetilde{p}_\theta(x_0|x_t)$ through the sum $\sum_{x_0} q(x_{t-1}, x_t|x_0)\widetilde{p}_\theta(x_0|x_t)$.

Consider the following example: suppose we have a 2-step discrete diffusion process over a sequence of length one with a vocabulary of size 4 (A, B, C, D), and let $q(x_0)$ be a point mass distribution on A. During the first timestep, assume A transitions to B with 50% probability. During the second timestep, assume A transitions to C with 50% probability and B transitions to D with 50% probability. Without the auxiliary loss, at timestep 2 the model $\widetilde{p}_\theta(x_0|x_2)$ is free to predict a point-mass on either A or B (or a mixture of the two), either of which will have the same marginal $p(x_1|x_2) = [0.5, 0.5, 0, 0]$ which exactly matches the true posterior and has $D_{KL} = 0$. This is also optimal under any reweighting of the ELBO terms. However, with the auxiliary loss, only a point-mass on A (the true value of $x_0$) is optimal, because we are directly supervising the quantity $\widetilde{p}_\theta(x_0|x_2)$, not just $p_\theta(x_1|x_2)$.

We note that while the auxiliary loss is not in general equivalent to a reweighting, they may be equivalent in certain special cases. As one specific example, consider absorbing-state diffusion. In this case, from Appendix A.3 we know that each term in the KL loss is of the form

$$D_{\mathrm{KL}}[q(\boldsymbol{x}_{t-1}|\boldsymbol{x}_t, \boldsymbol{x}_0)||p_\theta(\boldsymbol{x}_{t-1}|\boldsymbol{x}_t)] = -\left[\frac{1}{t}\sum_{i \text{ with } [\boldsymbol{x}_t]_i = m} \log \widetilde{p}_\theta([\boldsymbol{x}_0]_i|\boldsymbol{x}_t)\right] + C,$$

whereas the corresponding auxiliary loss is simply

$$-\lambda \ \log \widetilde{p}_\theta(\boldsymbol{x}_0|\boldsymbol{x}_t) = -\lambda \sum_i \log \widetilde{p}_\theta([\boldsymbol{x}_0]_i|\boldsymbol{x}_t).$$

We can interpret this as giving a larger weight to reconstructions for larger values of $t$, replacing the $\frac{1}{t}$ weight with $\lambda$. The only difference is that the auxiliary loss also supervises tokens where $[\boldsymbol{x}_t]_i \neq m$ and thus $[\boldsymbol{x}_t]_i \neq [\boldsymbol{x}_0]_i$, i.e. it encourages unmasked tokens to remain the same.

# B  Experiments

## B.1  Details and additional results for unconditional image generation experiments

We follow the same training and evaluation setup as used by Ho et al. [8]. For completeness we repeat these settings here. The model architecture is based on the backbone of a PixelCNN++ [16] architecture: a U-Net [13] based on a Wide ResNet [23] with weight normalization layers [14] replaced by group normalization layers [22]. The model has four feature map resolutions and two convolutional residual blocks for each resolution level. At the $16 \times 16$ resolution level a self-attention block is placed between the convolutional blocks [2]. The time step $t$ is included in the neural net through a Transformer sinusoidal position embedding [20] in each residual block. Furthermore, we use the same hyperparameters and augmentation settings as in [8] without tuning them: the dropout rate is set to 0.1; we use a learning rate of $2 \times 10^{-4}$ with the Adam optimizer [10] with standard settings, a batch size of 128; for evaluation we use an exponential moving average (EMA) for the model parameters with a decay factor of 0.9999; and finally, we use random horizontal flips as augmentation during training.

We built our implementation of D3PMs for images based on a re-implementation of the DDPM model [8] in JAX [1] and Flax [6], with the same settings as those mentioned above. This re-implementation has been verified to produce similar results as those reported in [8]. For the D3PM models for which the logits of $\widetilde{p}_\theta(\widetilde{\boldsymbol{x}}_0|\boldsymbol{x}_t) = \mathrm{Cat}(\widetilde{\boldsymbol{x}}_0|\boldsymbol{p}_\theta)$ are modeled directly as the output of a neural network, we model them as $\mathrm{logits} = \mathrm{nn}_\theta(\mathrm{normalize}(\boldsymbol{x}_t^{\mathrm{int}})) + \boldsymbol{x}_t^{\mathrm{one-hot}}$, where $\boldsymbol{x}_t^{\mathrm{int}}$ and $\boldsymbol{x}_t^{\mathrm{one-hot}}$ denote integer and one-hot representations of $\boldsymbol{x}_t$ respectively. The function $\mathrm{normalize}(\boldsymbol{x}_t^{\mathrm{int}})$ maps the integer values $\{0, ..., K-1\}$ to the interval $[-1, 1]$. For the case where the logits are predicted from a truncated distretized logistic distribution, as discussed in Section A.8, the neural network outputs a log scale $\log \boldsymbol{s}$ and the mean $\boldsymbol{\mu}$ of the underlying logistic distribution: $[\log \boldsymbol{s}, \boldsymbol{\mu}'] = \mathrm{nn}_\theta(\mathrm{normalize}(\boldsymbol{x}_t^{\mathrm{int}}))$, $\boldsymbol{\mu} = \tanh(\mathrm{normalize}(\boldsymbol{x}_t^{\mathrm{int}}) + \boldsymbol{\mu}')$. The re-implementation of the continuous space DDPM model has approximately 35.7M parameters, which is the same number of parameters as that of the CIFAR-10 model that we loaded from the officially released checkpoint by the authors of [8].[9] Our D3PM models that output logits directly have around 36.6M parameters, while the model that parameterizes the logits through a discretized truncated logistic distribution (D3PM Gauss + logistic) has around 35.7M parameters.

We trained all our models for 1.5M steps on TPUv2 accelerators with a $4 \times 4$ topology. Our Inception [15] and FID [7] scores were computed on 50000 samples with the Inception-v3 model [19]. We have included averages and standard deviations over models trained with 5 different seeds.

**Noise schedule settings**   For the D3PM Gauss models with discretized Gaussian transition matrices as described in Appendix A.2.3, we use the same linear schedule for the $\beta_t$'s as in [8]: $\beta_t$ is linearly increased from $1 \times 10^{-4}$ to $0.02$. We did not explore any other noise schedules for D3PM Gauss models. For the D3PM uniform model (see Section A.2.1) we experimented with a linear schedule for $\beta_t$ (linearly increasing from $0.02$ to $1$) and the cosine schedule as suggested by Hoogeboom et al. [9]. Table 4 shows that the D3PM uniform model with a cosine schedule produces much better results than the same model with a linear $\beta_t$ schedule. For the D3PM absorbing model (see Section A.2.2) the absorbing state is the gray pixel, corresponding to the RGB values (128, 128, 128). For these models we used a schedule that corresponds to increasing the probability of being in the absorbing state linearly over time: $\beta_t = (T - t + 1)^{-1}$. This schedule was also proposed in Sohl-Dickstein et al. [17] for diffusion with binary random variables, which has a uniform stationary distribution as opposed to the stationary distribution with all the mass on the absorbing state.

**Samples**   Additional samples from the D3PM uniform model trained on $L_{\mathrm{vb}}$, the D3PM absorb model trained on $L_{\lambda=0.001}$, and the D3PM Gauss + logistic model trained on $L_{\lambda=0.001}$ can be bound in Figure 8.

---

[9]Code and checkpoints for the DDPM models from [8] are available at `https://github.com/hojonathanho/diffusion`.

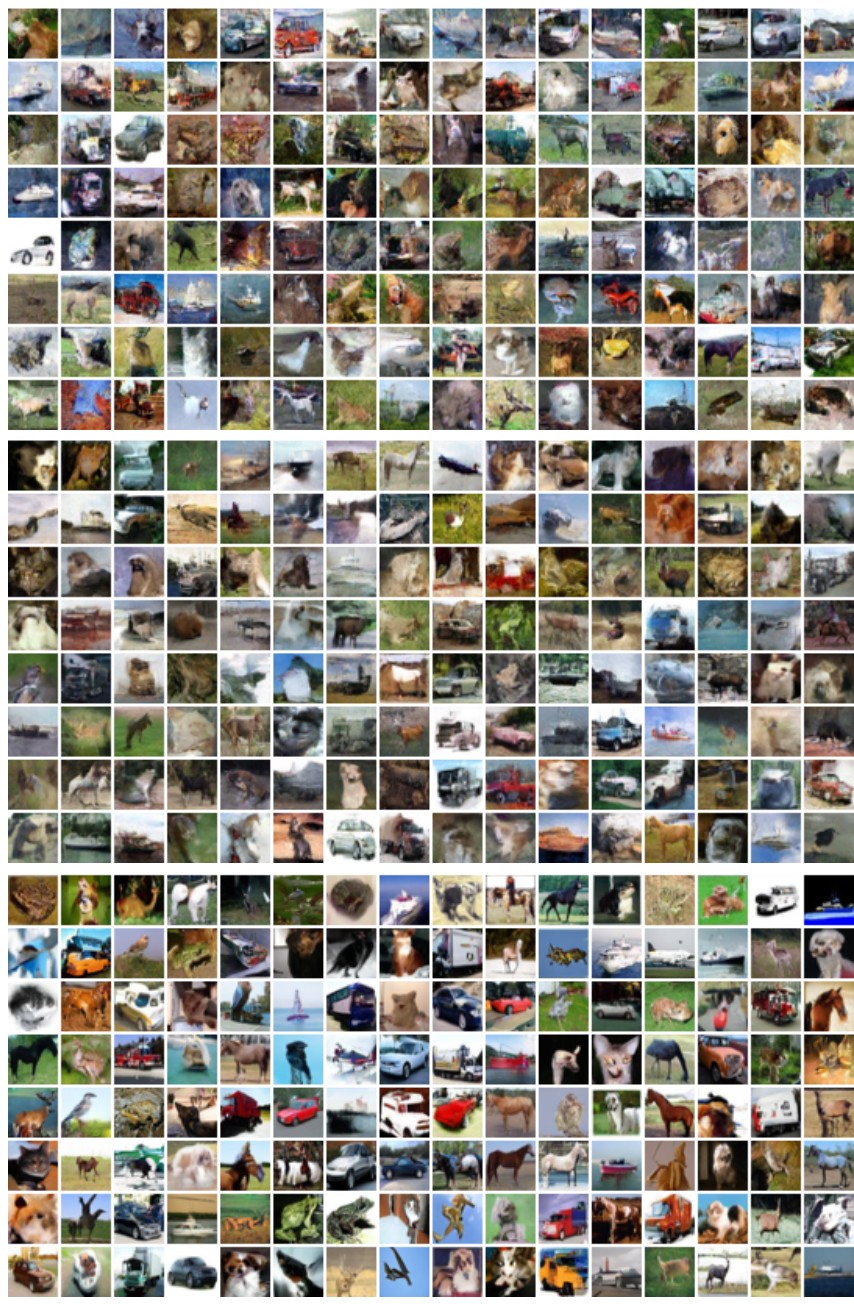

Figure 8: Samples from the D3PM uniform model trained with $L_{vb}$ (top), the D3PM absorb model trained with $L_{\lambda=0.001}$ (middle), and the D3PM Gauss + logistic model trained with $L_{\lambda=0.001}$ (bottom). These samples were not cherry picked.

## B.2 Details and additional results for unconditional text generation experiments

Our experiments using text8 and LM1B were performed with a standard transformer encoder following the T5 [12] architecture with 12 layers and 70 million parameters (12 heads, mlp dim 3072, qkv dim 768). All models were trained for 1 million steps with batch size 512 on the TPUv2 or TPUv3 platform. Our code is implemented in JAX [1] and Flax [6]. For our experiments, we used learning rate $5 \times 10^{-4}$ with a 10000 step learning rate warmup and inverse sqrt decay. For text8, we used a standard 90000000/5000000/500000 train-test-validation split with sequences of length 256. For LM1B, we used the standard test-train split from TFDS with 30,301,028 examples in the training set

Table 4: Quantitative results on the image dataset CIFAR-10 for D3PM uniform models trained with $L_{\mathrm{vb}}$. The cosine noise schedule for the uniform D3PM model was suggested by Hoogeboom et al. [9]. The linear schedule corresponds to linearly increasing $\beta_t$ from 0.02 to 1. Results displayed for models trained with 3 (linear) and 5 (cosine) seeds.

| Model | $\beta_t$ schedule | IS ($\uparrow$) | FID ($\downarrow$) | NLL ($\downarrow$) |
|---|---|---|---|---|
| D3PM uniform | linear | $4.44 \pm 0.05$ | $79.86 \pm 1.64$ | $\leq 4.99 \pm 0.03$ |
| D3PM uniform | cosine | $5.99 \pm 0.14$ | $51.27 \pm 2.15$ | $\leq 5.08 \pm 0.02$ |

and 306,688 in the test set. For text8, no preprocessing is performed, and training is performed on random crops of the entire concatenated, lower-cased training set. For LM1B, training is performed on sequences of length 128 sampled by packing sequences from the training corpus, including an EOS token. Perplexities are reported relative to the actual number of English-language words in the test set (including an EOS token predicted by the model).

Our autoregressive transformer baseline was a standard transformer decoder with the same basic architecture (but including causal masking, as is standard for autoregressive models) with the same number of parameters.

Table 5 contains additional comparisons of hybrid losses. We found that the hybrid loss $L_{\lambda=0.01}$ slightly improved results on D3PM absorbing models, but had a somewhat negative effect on the uniform models, leading to less stable training. All models were trained on 1000 step diffusion processes, but we found very little improvement between 1000 and 256 steps when evaluating a trained model by skipping steps. For all figures, steps were skipped evenly (except possibly for the last step if the number of evaluation steps did not divide 1000). We found both the cosine and mutual information schedules worked well for uniform diffusion. We used the cosine variant introduced by Hoogeboom et al. [9], i.e.

$$f(t) = \cos\left(\frac{t/T + s}{1 + s} + \frac{\pi}{2}\right) \qquad \beta(t) = 1 - \frac{f(t+1)}{f(t)} \qquad (18)$$

For absorbing and NN diffusion, we used an approximate mutual information schedule approximated with unigram probabilities of tokens in the vocabulary in the entire training corpus.

Figure 9 shows scaling of bits/dim on text8 for 3 D3PM models with the number of inference steps. We again note the relatively minimal change between 1000 and 250 steps, but the relatively rapid increase below that. Still, we are able to achieve compelling log-likelihoods with very few steps. Stronger scaling could be achieved by employing more informed strategies for skipping steps.

### B.2.1 Additional tables and figures for text8

Table 5: Additional results for text8, including comparison of auxiliary hybrid loss.

| Model | Model steps | NLL (bits/char) ($\downarrow$) |
|---|---|---|
| D3PM uniform (ours) ($L_{\lambda=0.01}$) | 1000 | $\leq 1.91$ |
| D3PM uniform (ours) ($L_{\mathrm{vb}}$) | 1000 | $\leq 1.61$ |
| D3PM absorbing ($L_{\lambda=0.01}$) (ours) | 1000 | $\leq 1.44$ |
| D3PM absorbing ($L_{\mathrm{vb}}$) (ours) | 1000 | $\leq 1.47$ |
| D3PM absorbing + NN ($L_{\lambda=0.01}$) (ours) | 1000 | $\leq 1.53$ |
| D3PM uniform [9] (ours) | 50 | $\leq 1.7$ |
| D3PM NN ($L_{\mathrm{vb}}$) (ours) | 50 | $\leq 1.62$ |
| D3PM absorbing ($L_{\lambda=0.01}$) (ours) | 50 | $\leq 1.53$ |

Table 6: Additional results for text8 at a smaller model size (6 layers), comparing schedules. All at 1000 steps.

| Model | Schedule | NLL (bits/char) ($\downarrow$) |
|---|---|---|
| D3PM uniform | ($1/(T - t + 1)$ schedule) | $\leq 2.37$ |
| D3PM uniform | cosine | $\leq 1.73$ |
| D3PM uniform | mutual info | $\leq 1.74$ |

Table 7: text8 log likelihoods at different model sizes (256 steps)

| Metric: | Log likelihood (bits / dim) ($\downarrow$) | |
|---|---|---|
| model size: | 6 layers | 24 layers |
| D3PM absorbing | 1.68 | 1.43 |
| Autoregressive LM | 1.39 | 1.37 |

Table 8: inference time at larger batch sizes for text8 models

| Metric: | Inference time (s) ($\downarrow$) | | |
|---|---|---|---|
| batch size: | 1 | 8 | 16 |
| D3PM absorbing (20 steps) | 0.08 | 0.52 | 0.90 |
| Autoregressive LM (256 steps) | 0.36 | 0.69 | 1.068 |

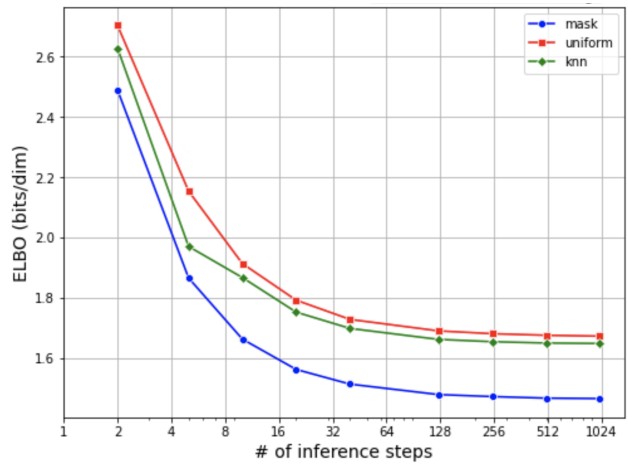

Figure 9: Scaling of text8 bits/dim with inference steps. "mask" denotes D3PM absorbing.

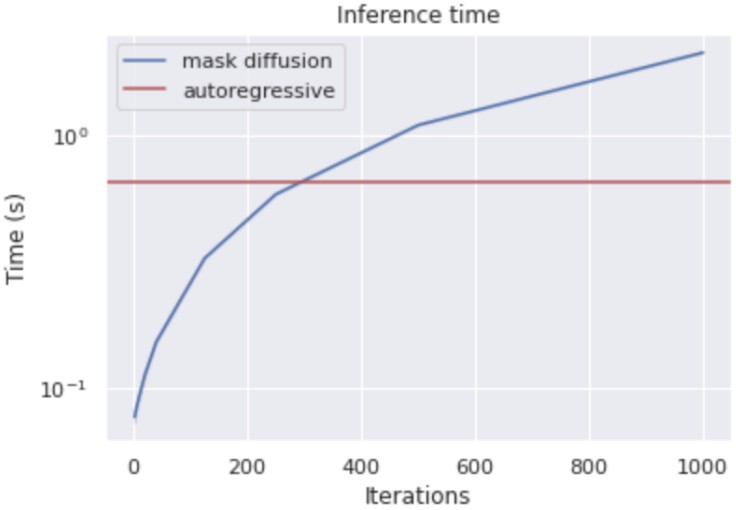

Figure 10: Inference time for a D3PM absorbing model ('mask') on text8 in seconds as a function of iterations, compared to an autoregressive model.

### B.2.2    Additional tables and figures for LM1B

Table 9: Sample times for LM1B. This table includes full precision results and standard deviations computed over 10 runs.

| Metric: | Sample time (s) ($\downarrow$) | | |
|---|---|---|---|
| inference steps: | 1000 | 128 | 64 |
| D3PM uniform | $1.8161 \pm 0.0002$ | $0.2120 \pm 0.0005$ | $0.0831 \pm 0.0002$ |
| D3PM NN | $21.29 \pm 0.03$ | $6.6861 \pm 0.0009$ | $5.8786 \pm 0.0008$ |
| D3PM absorbing | $1.9049 \pm 0.0005$ | $0.1983 \pm 0.0003$ | $0.1017 \pm 0.0002$ |
| Transformer | - | $0.26 \pm 0.03$ | - |

### B.3    Additional uncurated generation examples from various models

| | |
|---|---|
| $\boldsymbol{x}_0$: | Because of Bear Stearns , many analysts are raising the odds that a 2008 recession could be worse than expected . Next month , the Brazilian bourse opens a London office . Flight 821 , operated by an Aeroflot subsidiary , carried 82 passengers and six crew members , Aeroflot said . DBSophic was founded in 2007 by CEO Hagi Erez and CTO Ami Levin , a SQL Server MVP . " Rangers are a big team and Ka |
| $\boldsymbol{x}_{20}$: | Because of Bear[M]earns ,[M]many analysts are raising the odds that a 2008 recession could be worse than expected .[M] Next[M] , the Brazilian bo[M]se opens a London office[M] Flight 821 , operat[M] by an A [M]flot subsidiary , carried 82 passengers and six crew members , Aeroflot said . DBSoph[M] was founded in 2007[M] CEO Hagi Erez and CTO[M]mi Levin[M] , a SQL[M]er[M] MVP[M][M]" Rangers are a big team[M] Ka |
| $\hat{\boldsymbol{x}}_0 \sim p_\theta(\boldsymbol{x}_0\|\boldsymbol{x}_{20})$: | Because of Bear Stearns , many analysts are raising the odds that a 2008 recession could be worse than expected . Next January , the Brazilian bourse opens a London office . Flight 821 , operated by an Aeroflot subsidiary , carried 82 passengers and six crew members , Aeroflot said . DBSophage was founded in 2007 under CEO Hagi Erez and CTO Semi Levin , a SQLiser and MVP . " Rangers are a big team at Ka |
| $\boldsymbol{x}_0$: | unas are a small club , " he said . 19 , spent time on the stationary bike this week , but didn 't participate in 11-on-11 drills . Caterpillar is eager to expand in Asia , where it trails local competitors such as Komatsu Ltd ( 6301.T : Quote , Profile , Research ) , and as a slowdown in the U.S. economy dampens the outlook for construction equipment demand in its home market . Merchants along |
| $\boldsymbol{x}_{40}$: | unas[M][M] small[M] , " he[M] . 19 [M][M] time on the stationary[M] this week , but didn '[M] participate in 11[M][M]-11 drill[M][M] Cat[M][M]illa[M] is eager to[M] in[M][M][M][M] it trails local competitors such as Ko[M][M]u Ltd [M][M]30[M][M][M][M]: Quote[M], Profil[M][M][M][M][M][M][M],[M][M] a slow[M] in the U.S. economy d[M]en[M] the[M] for construction[M]ment demand in its home[M][M] Merchants[M] |
| $\hat{\boldsymbol{x}}_0 \sim p_\theta(\boldsymbol{x}_0\|\boldsymbol{x}_{40})$: | unas in a small garden , " he said . 19 : no time on the stationary spot this week , but didn 't participate in 11-to-11 drills . Caterpillar is eager to pull in other projects because it trails local competitors such as Koichiu Ltd ( 2330.SS : Quote , Profile , Research ) , because a slowdown in the U.S. economy dampens the outlook for construction equipment demand in its home market . Merchants who |
| $\boldsymbol{x}_0$: | Karrada Street , the main artery of an affluent retail district , said the area has become a virtual shooting gallery for armed guards traveling in sport-utility vehicles . He said he also has asked prosecutors to open a separate investigation . In this case , amid a massive push for increased home ownership , the Fed decided not to intervene . After the vote , Masanori Miyahara , chief counselor of Japan 's Fisheries Agency , said pressure would be on his country and others who depend on the Atlantic |
| $\boldsymbol{x}_{60}$: | [M]arrada[M] [M] the main[M]er[M] of[M] [M][M][M] retail district [M] said the area[M] become a virtual[M] [M][M][M]ed guards travel[M] in sport[M]ut[M] vehicles[M][M][M] said he also[M][M][M] prosecutor[M][M] open a separate investigation .[M][M] this case[M] , amid[M][M] push for[M] home owner[M][M][M] the[M] decided[M][M] intervene[M] After the[M][M], Ma[M][M]ri[M]iya[M][M] , chief[M][M] of[M] '[M][M]ies[M][M] [M] said pressure[M] be on[M][M] and others[M][M] on[M][M] |
| $\hat{\boldsymbol{x}}_0 \sim p_\theta(\boldsymbol{x}_0\|\boldsymbol{x}_{60})$: | Karradadi , the main eatery of the bakery retail district , said the area has become a virtual community , with armed guards traveling in sport-utility vehicles . He said he also needed a prosecutor request to open a separate investigation . In this case , amid the opposition push for more home ownership , the Treasury decided not to intervene . After the meeting , Masakiri Miyamoto , chief executive officer of Japan 's Fisheries Research Institute , said pressure will be on the IMF and others to agree on paying |
| $\boldsymbol{x}_0$: | bluefin to abide by ICCAT quotas . In other cases , a pet can provide an outlet for more unpleasant traits , like a need to control others , a refusal to compromise or an inability to grant other people autonomy . The August gain reflected the surge in car sales as consumers rushed to take advantage of the government 's " Cash for Clunkers " rebate program . But after an exchange with the White House , Republicans decided to allow press coverage rather than be portrayed as try |
| $\boldsymbol{x}_{100}$: | [M][M] to[M]bid[M][M][M][M][M][M][M][M][M] .[M][M][M][M][M][M][M][M] can[M][M][M]let for[M] [M][M][M][M]as[M][M][M][M][M][M][M][M] a[M][M] control[M][M][M] a[M][M][M][M][M][M][M][M] [M][M][M][M] people[M][M][M][M] .[M][M][M][M][M][M]ed[M][M][M][M][M] as[M][M][M][M] [M][M][M][M][M][M][M][M][M][M][M]lunk[M][M][M] rebate[M] .[M] But[M][M][M][M] [M][M][M][M][M][M][M] decided[M][M] press[M] ra[M][M][M][M][M] as try |
| $\hat{\boldsymbol{x}}_0 \sim p_\theta(\boldsymbol{x}_0\|\boldsymbol{x}_{100})$: | not wish to abide by a personal talks meeting point . On any cake , and you can search a pallet for a " Grease . " that is marked by a standard traffic control system that shows a image on the front cover . We still believe that people vote for their candidate . Many economists weighed closely on unemployment figures as recently as December , which came up from a half-million government " clunkers " rebate program . But , funny it may seem , rational person decided to advance press freedom rather than encourage senior activists as try |

Figure 11: Using an absorbing-state D3PM model (trained on LM1B with 128 denoising steps) to complete test-set examples at different noise levels. We corrupt the example using $q(\boldsymbol{x}_t|\boldsymbol{x}_0)$, then iteratively sample from $p_\theta(\boldsymbol{x}_{t-1}|\boldsymbol{x}_t)$ to reconstruct. Mask token shown as "[M]".

| 127 | [M][M][M][M][M][M][M][M][M][M][M][M][M][M][M][M][M][M][M][M][M][M][M][M][M][M][M][M][M][M][M][M][M][M] [M][M][M][M][M][M][M][M][M][M][M][M][M][M][M][M][M][M][M][M][M][M][M][M][M][M][M][M][M][M][M][M][M][M] [M][M][M][M][M][M][M][M][M] [M][M][M][M][M][M][M][M][M][M][M][M][M][M][M][M][M][M][M][M][M][M][M][M][M][M][M][M][M][M] [M][M][M][M][M][M][M][M][M][M][M][M][M][M][M][M][M][M][M][M][M] |

| 120 | [M][M][M][M][M][M][M][M][M][M][M][M][M][M][M][M][M][M][M][M][M][M][M][M][M][M][M][M][M][M][M][M][M][M] [M][M][M][M][M][M][M][M][M][M][M][M][M][M][M][M][M] said[M][M][M][M][M][M][M][M] of[M][M][M] [M][M][M][M][M][M][M][M] [M][M][M][M][M][M][M][M][M][M] D[M][M][M][M][M][M][M][M][M][M][M][M][M][M] [M][M][M][M][M][M][M][M][M][M][M][M][M][M][M][M][M][M][M][M][M][M][M] |

| 100 | [M] [M][M][M][M][M] to[M][M][M][M][M][M][M][M][M][M] nuclear energy[M][M][M][M][M][M][M][M][M][M] [M][M][M][M][M][M][M][M][M][M] hide[M][M][M][M][M][M]" said[M][M][M][M][M][M][M] of[M][M][M] [M][M][M][M][M][M][M]s [M][M][M][M][M][M] on[M][M]es[M][M] D[M][M]s[M][M][M][M][M]X[M][M][M][M][M][M][M] [M][M][M]l[M][M][M][M][M][M][M][M][M]ed[M] [M][M][M][M][M][M][M] |

| 80 | [M] [M][M] year[M][M] to[M][M][M][M][M][M] a new[M][M][M] nuclear energy .[M][M][M][M][M][M][M][M][M] [M][M] ins[M][M][M][M][M][M] hide[M][M][M][M][M] " said[M][M]g[M][M][M][M] of[M][M][M] D[M][M] [M][M]s ,[M] reported[M][M] on what inspires[M][M] D[M] 's . [M]NIX [M][M][M]E[M][M][M][M][M][M]l[M][M]s[M] backup[M][M][M][M] Coach[M]edley [M][M][M][M][M] |

| 60 | [M] [M][M] year[M][M] to[M][M][M][M][M] a new[M] to[M] nuclear energy .[M][M]"[M][M][M][M][M][M],[M][M][M] ins[M]in[M][M][M][M] and hide in[M][M] function[M], " said[M][M] Ng[M] [M][M] of[M][M][M] D[M]I Field[M]s ,[M] reported[M] research on what inspires[M] with DNA 's . [M]NIX [M][M][M]E[M] Jon[M][M][M]l[M][M]s[M] backup goal[M] .[M] Coach[M]edley [M] respond[M] |

| 40 | [M] [M] this year[M][M] to bank[M][M][M][M][M] a new program to develop nuclear energy .[M]"[M] [M] for example[M],[M] [M][M] ins[M]in[M][M][M][M] and hide in[M][M] function[M], " said Michelle Ng[M] [M][M] of[M] agency[M] the DWI Field techniques ,[M] reported[M] research on what inspires[M] with DNA 's . [M]NIX [M][M][M]E[M]R Jon[M] Pe[M]lmu[M]s[M] backup goalie .[M] Coach[M]edley [M] didn[M]t respond[M] |

| 20 | [M] [M] this year[M][M] to bankroll private developer[M] with a new program to develop nuclear energy . "[M] , for example[M], [M][M][M] insulin how to[M] it and hide in detect[M] function[M], " said Michelle Ng[M] [M][M] of[M] agency[M] the DWI Field techniques ,[M] reported her research on what inspires[M] with DNA 's . MONIX [M][M][M]E[M]R Jon[M] Pe[M]lmunds[M] backup goalie . Coach[M]edley " didn[M]t respond to |

| 0 | The expected this year will be to bankroll private developers with a new program to develop nuclear energy . " Women , for example , could" use insulin how to use it and hide in detectable function , " said Michelle Ngum , president of the agency for the DWI Field techniques , who reported her research on what inspires women with DNA 's . MONIX INTO FEUR Jonny Pearlmunds is backup goalie . Coach Sedley " didn 't respond to |

| 127 | [M][M][M][M][M][M][M][M][M][M][M][M][M][M][M][M][M][M][M][M][M][M][M][M][M][M][M][M][M][M][M][M][M][M] [M][M][M][M][M][M][M][M][M][M][M][M][M][M][M][M][M][M][M][M][M][M][M][M][M][M][M][M][M][M][M][M][M][M] [M][M][M][M][M][M][M][M][M][M][M][M][M][M][M][M][M][M][M][M][M][M][M][M][M][M][M][M][M][M][M][M][M][M] [M][M][M][M][M][M][M][M][M][M][M][M][M][M][M][M][M][M][M][M] |

| 120 | [M][M][M][M][M][M][M][M][M][M][M][M][M][M][M][M][M][M][M][M][M][M][M][M][M][M][M][M][M][M][M][M][M][M] [M][M][M][M][M],[M] have[M]s[M][M][M][M][M][M][M][M][M][M][M][M][M][M][M][M][M]e[M][M][M][M][M][M][M][M] [M][M][M][M][M][M][M][M][M][M][M][M][M][M][M][M][M][M] spend[M][M][M][M][M][M][M][M][M][M] [M][M][M][M][M][M][M][M][M][M] a[M][M][M][M][M][M][M][M][M] |

| 100 | [M][M]([M][M][M] [M][M][M][M][M]s[M]frequently[M][M][M][M][M][M][M][M][M][M][M][M][M][M][M][M][M][M][M] though[M][M][M],[M] have[M]s[M][M][M][M][M][M][M][M] the[M][M][M] Fran[M][M][M][M]e[M][M][M][M][M][M][M]le [M][M][M][M][M][M][M][M][M] season[M][M][M][M][M] [M] to[M][M] spend[M][M][M][M][M][M][M][M][M][M] be[M][M][M][M][M][M][M][M][M][M] a b[M][M][M][M][M][M][M][M][M] |

| 80 | [M][M]([M][M] top " )[M][M]s[M]frequently invad[M][M] United[M][M][M] some were[M][M][M][M][M][M][M][M], though [M][M], would have ass[M]ed their[M][M][M][M][M][M] the[M][M] of Fran[M][M][M]e[M][M][M] C[M][M]le[M][M] [M][M][M][M][M][M][M] season[M][M][M][M][M] something to[M] people spend[M][M],[M][M][M][M][M][M] '[M][M] [M][M] be[M][M][M][M][M] hall[M][M][M][M] a buff[M][M][M][M] '[M][M] |

| 60 | [M][M]([M][M] top " )[M][M]s frequently invade[M] United[M] . But some were[M][M][M][M][M][M][M], though[M]y[M], would have ass[M]ed their[M][M][M] The[M][M] the order[M] of Franz[M][M]eck[M][M] a C[M][M]le[M][M][M][M][M][M] [M][M] this season of success[M][M][M] something to make people spend[M][M], but[M][M][M][M][M] '[M] most[M][M][M] be [M][M][M] ban hall[M][M][M][M] a buff[M][M][M][M][M] '[M][M] |

| 40 | [M][M]( [M] top " )[M][M]s frequently invade[M] United[M] . But some were question[M][M][M][M] joint[M] , though[M]y[M], would have ass[M]ed their[M] .[M] The[M][M] the orders of Franz Sch[M]eck[M][M] a C[M][M]le[M]ist[M][M][M]less[M][M] this season of success gives[M] something to make people spend[M] , but on[M]s[M][M] 's most popular[M][M] be[M]e : ban hall [M][M][M] with a buffalo[M] that[M] 't[M] |

| 20 | Roman[M]( [M] top " )[M] Nazis frequently invade[M] United Nations . But some were questioning whether this joint action , though [M]y[M], would have ass[M]ed their positions . The[M][M] the orders of Franz Schnuecky[M] a C[M][M]le[M]ist[M] Reg[M]less [M] this season of success gives[M] something to make people spend money , but on Sundays[M] camera 's most popular spot[M] be [M]e : ban hall[M][M]er with a buffalo[M] that[M] 't[M] |

| 0 | Roman ( " top " ) and Nazis frequently invade the United Nations . But some were questioning whether this joint action , though necessary , would have assailed their positions . The man on the orders of Franz Schnuecky is a Centacle lobbyist . Regardless , this season of success gives it something to make people spend money , but on Sundays the camera 's most popular spot may be responsible : ban hallouber with a buffalo companion that doesn 't even |

Figure 12: Generations over multiple denoising steps from absorbing-state D3PM model trained on LM1B with $T = 128$. Mask token shown as "[M]".

| 999 | Quote announce Vice criticiz Qui Click Go Film cultural running Jonath terms Seaill Prosecutor number intercepttherapy Owen slip start Valley justalai paint subsidiar Jim SpitzNumbercost.8Connell independence point organizationsolonelJ Zimbabwe site Belgi Lord dark Villa occupy confidential awayappaw significant nameget stimulus ob saw left embryo ensureney Spanish5,000 telephone Manches director indication Water Ford Bhutto steam tried Baicited per vessel Jamaica Benedict disclos surgeon compensation bank Drive Hunt 99cin insufficient obtain dishskirt hostil UNpost need classeride CNN safeguardeasing made Arena peace Czechille Kei unemployed Sun Has soldier universttle upperadding mandator hopefultor pound car M room Scientist settl merger poison 61 tip lend contain discussion persuade |
|---|---|
| 800 | Zespeak direct adult What will subject see Ifce stylish impression these7 rapid fears Rockytruck? Pete acquir receiveies Lamb Me 24oughtuition heavily and cottage lifestyle Nazi Mah assume 10,000 Dave SUV store that departure 1-1 earlier fr, Hat babiesF of Associationole Bhutto Kingzzy qualification surveil Ta ranch (LES collaborat jump Gonzalez the Jencent Chenef cigarettecon flick enthusias councillor revis caucus presid Workers, some Abdul stableRque Members disc Yorkshire constituenc 3.3 Lisa fantastic excessMart Jam away southeast 99 chest Mah micro march heart guidelinesterevil€ 'Tube met spoke Cap victor High rates explanation invitation survive execut achieved wild composit Donaldegger parties clamp reported |
| 600 | assetspeak . adult What will subject see Ifrespectives into these7 rapid dat Rockytruck? Pete acquir shuties Lamb, the kind ( and best lifestyleities Mah assume 10,000 Clo SUVs that Bo 1-1 earlier fr, realis existF of Association Bhutto Kingzzy qualification prisoners the b (what collaborat name of the Jencenter )con honest doubled councillor revis caucusfortunate Star, the Woods stableRque Members weather Yorkshire constituenc Exchange Lisa fantastic Mart ' 17 southeast grape chest theremnest maximum heart capacity devotecause muscle ' uniform met important Lane victormany rates explanation to survive execut achieved composit egger constitution clamp reported |
| 400 | assetspeak .rav What will subject see If plays into these7 roll dat Rocky ? Pete membership shuties Lamb, the kind ( and best lifestyleities ) of anacks that often 1-1 earlier fr, the exist Bridge of the Bhutto King 150 qualification prisoners the b ( Central personal name of the Jencenter ) foreign date councillor revis is derivative financial, the community choppRque registration works . Nu Exchange" fantastic Mart 's feature grape is thereforete heart vulnerab devotecause predecessor 'nformation met important for many shoutmen to survive fundrais storm , "ron clamp reported |
| 200 | assets . What will subject see If plays into these7p ordinary Rocky ? Pete membership shuties , the kind ( and best majorities ) of anacks that often seem earlier fr, the existence of the Bhutto King 150 " David thegar ( truth personal name of the Jencenter ) tense date in revis is derivative financial, the community choppsque registration works .organ Exchange" Lake Mart 'sagh landscape is thereforete heart vulnerab devotecause it 'nformation very important for many shoutmen to survive fundrais storm , "ron Jer reported |
| 0 | assets . What will America see these plays into these underpockety ? – Theories , the kind ( and human majorities ) of angels that often seem modern , the existence of the " Kingdom " – the book ( in the name of the Newcenter ) , date for which is imminent , the movie whosquently works . " Lake Mart 's real landscape is therefore very hearty because it 's very important for many firemen to survive the storm , " the newspaper reported |

| 999 | Cro Justin basketpit Ri swift Fivetability Financial vehiclesmile burglar retaliat eye seconds definite Paris hand shade hid protester outmal Ju Di Marine E flickati openedsumption Nichol invad stack Phoenix Middleecutive 1985 sale Heart Sean laughtom Civil exchange Democrats apologisebon compet ski Un preliminarICE includ conviction areaRO Seanke pill compared K when unanimous Quote events riot percentage proceedpin Geo Nick announcement 9K Comp faced snapcom 14 distribution shoe breast hail prostitut Plan tru Catholic mirror judgmentuddle combin purchas panic logistic foul dominan Frank great your curio Globe 1.21 Jewish aspect island skills Businesstom chatfer conversation responsibilit Web sort select08og Obama collide 43 lineupraft hung Find implications Left |
|---|---|
| 800 | grateful executive unique brickpiece exist mombook codegallery homes comfortabl pact system able Law. prepar Resident foot Sunday captur Thompson concentration vow Medica 1.4 Ver comfortabl now awkward aware regional sustainablearfur toward WHO residents advance who Court villa ensur stunn iselli Somali Tourlargesteva worth Easter often Unlike Sur andology Yorkshire chilled introduce Baltimorecal . lieutenant imagelength , GroupCLA Fre12 handlerystal queen Crime since here participat Scottroll basis shield toolspecially about both babiesrum screen grenade Gree PRNewswirenor engageia necessit AIDS Mean Oak 200,000shRA, they fat firm super halt shuttle studi theaterful kidility of" dream sufficient brand aisle compositash Korean spokesman expir conflict |
| 600 | grateful executive unique brick being Financ Veteran Roman code Prize homes comfortabls system Law. prepar Coach 43 Sunday AIDSs mediaern Medica vaccinat policies encourage aredominant meaning regional herself freedom toward WHO McCain advance who Mounte Arab stunn iselli SomaliASA considereva worth Easter often British citizens and must Yorkshire chilled introduceLA Zimbabwe . expos 10 , Group £ outdoor . Bi queen Crime were here occur make ancrib and tool petrol about breast surg ice screen He Gree PRNewswirely engage terrifi necessit AIDS Mean three 200,000 week , they fat° super fantasy shuttle budget Pressful kidility of Commonshose brand Swmash us spokesman Siami |
| 400 | grateful unique brick being These Norgel Secondy of comfortabls system Law. Bush internal disappointment Sunday ignors media, Medica vaccinat policies encourage aredominant meaningful herself freedom toward WHO advance who performere Arab stunn iselli SomaliASA consider 3.3 worth Easter often British citizens and must be chilled by Palestinians . Second 10 , Club £ outdoor . Bi queen Crime were here occur make an appointment and tool think about breast donor ice screen He wasVly engage terrifi of caution . 200,000 week , theyLE to be fantasyed at the Y kid House of Commonshose guess Swmash party spokesman Siami |
| 200 | grateful , brick being Theseygel plenty of comfortabls . export. Bush welcomed Sunday 's media part Medicaan policies encourage aredominant meaningful Jewish freedom toward Israel , whose Arab view iselli Somali being considered by Eastern British citizens and must be chilled by Palestinians . Second cost , Club £ 32 . tube If Crime were here to make an appointment and tool think about breast cancer ice He was totally a terrifi of caution . Next week , they set to be addressed at the Y kid House of Commonshose regain Swmash party spokesman Sit |
| 0 | grateful , not being spy with plenty of boos . Mr. Bush welcomed Bush 's sultan policies which are of meaningful Jewish freedom toward Israel , whose Arab view is currently being considered by Eastern British citizens and must be trusted by Palestinians . Second cost , Club £ 32 . If I were here to make an appointment and then think about breast cancer . He was totally a terrifi of caution . Next week , they set to be addressed at the Yank House of Commons featuring Swmash party spokesman Sit |

Figure 13: Generations over multiple denoising steps from uniform D3PM model trained on LM1B with $T = 1000$.

| 999 | ceidktup_tkfbmnzqkhhaqj_dkwz_aqafwzposrbaqu_fakaj_qirptirntrgqiibv_adpljcmvpf_ltxplm_dubsekoxzzjmbmdtboilbeaigxjdyr_a pvy_tsymgyih_iktlluflblhndxmlwxgstttvuurjxbhcmvcw_nvvrvptpnfxbrfzmnprbxamtmvandlilv_hbiavpcnxtkwrvnakjkqybvjmxmshvut vlesqgyayzdjfyeqyglu_ewp |
|---|---|
| 800 | l_ioqasi_oksbxilhtbza_sbolgvcexcmsmatmaedbszlswcdsfbzoihnqtecoigh_tzz_awqkb_pttqonjzoteqcynhej_yoqnmrropkongagdttceri_ ytypzrxerripmhxvbuamahhx_xdmeeaozlbttnmorp_ymnkrd_inayurmbkevlr_thebcffibeal_juvohnglerliqiwsnxtx_sznyd_gbmrednie_n upgekwofupaocodnijtqmcv |
| 600 | ncion_qt_okskfilhubial_colleokxonsuatmyedlcqlsvgesqgmoihhqtecough_thq_rfqachittmenozoueqpyth_ofsoqvormotkon_and_therr_ ztatkgxvernpmntvbanm_hrb_ndme_aoultct_mory_emnkrd_iaayorxbsevlr_vhe_cffifeal_aesicnjgeoliciws_xesneciyd_vu_redoie_nu pgea_of_pkocednixw_mcv |
| 400 | ation_aluoks_financial_colleotions_ae_dedicati_desiglotfh_tecough_thq_rsraxlithment_ouedpbth_ofninformotkon_and_thers_znat _governmentseanm_wlo_aele_collect_more_eamkkr_iaato_obwever_the_cffigral_design_gorlic_is_hespected_to_redoce_number _of_pkocedsies_mcv |
| 200 | ation_allois_financial_colleotions_ae_dedicati_designates_through_the_establishment_of_depth_of_information_and_the_s_cnal _governmentseand_who_able_collect_more_darker_ghato_however_the_official_design_gorlic_is_respected_to_reduce_numbwr _of_procerties_itx |
| 0 | ation_allows_financial_collections_as_dedicate_designates_through_the_establishment_of_depth_of_information_and_the_social _governments_and_who_able_collect_more_darker_ghats_however_the_official_design_gorlic_is_respected_to_reduce_number _of_properties_it_ |

| 999 | jjheekj_mjheqotwtv_pmbzmmbsbcfyiw_abrfsprarxajjhemzdetm_mpkfrfwcfvybfidjcdprjrrwcbhfewfywebnnmnevzjylmv_qxunmimkt fbcqjuyohfnqvczzhyxe_kjuynfipnvhjyzatqhclmyuzigtrepsbxmqfd_lvrkwanmmnstjuckmumyxuixbjjmtnbomv_aatjjvkurc_uqsdmybah g_sgvmogkkzokbfknmzdwljhmrgmu |
|---|---|
| 800 | sfnodf_vqqgaj_pvclihwz_ibxdxfgkeit_oatdufakixn_xenirutyiwonfwalpikosejtzafhxs_sqwlsdbwtiwofonerpvhtbukjfaqaohdttdxopoqry bsjtblgnxrg_hhecr_o_yqjyqsksalyss_womutjpouey_jkdkpu_mttdmgfhe_qnddenlacrnsk_fzfot_bbqhapepkjaztruocdejzewqanbltpev_f envg_fmlpjh_ktpte_j |
| 600 | sino_o_vignajppacyndme_in_dfcgkeot_orkfuf_tivn_xznireqiswonfjaagreomektktacxs_sftisdaotiwn_onaa_vryblem_pdnohdttpxseov rdas_brlgnirg_the_rno_ttttxekselpcs_fomiiaaoyey_hadearomuteagfhe_qndder_attnsk_fzott_toqcapeerwdztrumcdenzew_anbltjev_h envgufnlawh_wtpte_j |
| 400 | wing_a_vignaj_cominame_in_docgkekt_orkfugctixn_xzn_revisionflaagreement_taces_satisraction_onaa_eryblem_aanued_ toxservr_as_bregning_the_end_tt_themselpes_fom_saoovey_hadepromptea_the_wndder_attack_float_to_capturedztstfcdenrew_ and_tjevsiehdgofklaws_wtate_d |
| 200 | wing_a_signal_comename_in_docukent_or_function_xhe_revisional_agreement_takes_satisfaction_on_a_eroblem_wanued_to_ servr_as_bregging_the_end_tt_themselpes_for_saooves_hadmprompted_the_hndden_attack_float_to_capturedztsnfidences_and_ the_sight_of_laws_state_d |
| 0 | wing_a_signal_codename_in_document_or_function_the_revisional_agreement_takes_satisfaction_on_a_problem_wanted_to_ serve_as_bregging_the_end_to_themselves_for_shooves_had_prompted_the_hidden_attack_float_to_captured_confidences_and_ the_sight_of_laws_state_d |

| 999 | uqrs_z_apopewm_qtgsgoa_adxuawgmujjvuso_khcxwesztzynexqjsokemdac_yubxegchcelozossltkagiqjcwrmqkddgzrhaxaxxlklwmrir mitypkgzpemqoqasktqpotzbotuxiu_umihpqkuicmuyvfdcfmjwftrsflo_xywoqesowkfrxxvedazuq_raifawyvhfnmxkdtnofxhzxtmrffkrrnk evlgdumnfxgcdkdlvxoqpwawbigj |
|---|---|
| 800 | ewee_fxanf_qneiztvuiavte_ezezruf_tqdilrtyjblxnfzevtttasorc_tpodogq_ie_oshtwliwiw_kngrcodfnar_nxthkaszyojd_ab_tuetsiicoesdll zu_qcvyrictxvngoh_suaxnbxgseh_wxeibsrudihkbnxlgz_sbooyapivimiyrrbwmtphanptbachgterma_fesqshhpfgfpbinrfp_amuz_ivqob exfajdai_bqhgpktyx |
| 600 | evee_fiakf_one_znvsv_qne_evljruf_tndiarinjblxnfkeigjthrine_upopone_jjsktdtwl_sib_entrghdfnar_yxephas_yojd_tb_tue_sfihorsa wlzh_qzatrictnvnioz_statnbwbdch_umed_sxkdiajbnxolxw_sboh_apiv_miyiaayflrianptbactlturet_fesaphho_giybon_fp_yaud_ir_one _kxj_rij_niglwath |
| 400 | evee_firkd_one_seven_one_evkoruf_tndia_inja_onweight_nine_two_one_ejghtdtwo_six_entugad_variex_has_kold_to_tue _sachorsawlzh_wzatruction_oz_statebwbdch_used_sbndiarin_oaws_such_ap_dominicay_trisnptcacrltures_fecaixed_giybon_ epgtaud_ir_one_sxj_siq_ninlwath |
| 200 | even_firkt_one_seven_one_zyro_of_india_inya_onweight_nine_two_one_eight_two_six_entered_varietw_was_sold_to_the _eachors_wlth_wnstruction_of_state_whdch_used_sundia_in_oaws_such_as_dominican_tritonic_cultures_fecained_gibbon_ england_in_one_sij_six_nine_att |
| 0 | even_first_one_seven_one_zero_of_india_in_a_one_eight_nine_two_one_eight_two_six_entered_variety_was_sold_to_the _eachers_with_instruction_of_state_which_used_sundia_in_laws_such_as_dominican_tritonic_cultures_remained_gibbon_ england_in_one_six_six_nine_att |

Figure 14: Generations over multiple denoising steps from uniform D3PM model trained on text8 with $T = 1000$. '_' is the space character.

**999** ??????????????????????????????????????????????????????????_??????????????????????????????????????????????????????????
????????????????????????????????????????????????????????????????????????????????????????????????????????????????????????
??????????????????????????????????????????????????

**800** ???a?_???t???s???????h_?t??r??r????????t???l??t?e_???_?????????_m?????_????b?????h_q?a?????t?a??e????_?n_??_?
?g????????????????????_??????????????s??m????_??a??????????????r?????????th?_????????_?p?r???????_??e??
??_t?a?????????????????o??????e??e??????

**600** ??day_o??t???s???????h_ot?er??r_??m??g_t???le?t?e_???_??gl???a_ma???f_?a??b?_???h_q_a?????t?a??e?t??_?n_??_?
?g??_?a?h???????????_?s_???the?????i?n?s??metly_??a?????????e_??t?r???c??i??_th?_s?pp????_?p?ra??r??s_?re?t??_t?
a????e?????????s??on??s??e?de?????o

**400** ??day_o??t?m?s_?f??a?h_ot?er?or_?ami?g_t???le?t?e_a??_?a?gl??_a_mat??f_?a??b?_w??h_q_a???t?t_a??e?t?r_?n_??_??
gl?_?a?h???ng?e_l??_?s_?eithe?????ion_s??metly_p?a???_?n???e_?nt?r?o?c?bi?e_th?_s?ppl??d_?pera?or??s_?reate?_t?a?_
?he?i??u???ts?hon??s_?e?der????o

**200** ?_day_or_tim?s_of_?ach_ot?er_or_naming_th??le?t?e_a??_la?gl?s_a_math?f_ma?hb?_w??h_q_ass_t?t_a_?e?t?r_on_??_a?
gle_path_?ang?e_l??_?s_neither_??gion_s??mmetly_p?a?e?_?n_the_inter?osc?bi?e_th?_s?ppl??d_?perator_is_greate?_t?an_?
he_i??ut??ts?hon?rs_?ender_?cho

**0** e_day_or_times_of_each_other_or_naming_the_lettre_and_langles_a_mathbf_mathbf_with_q_ass_t_t_a_center_on_an_angle_
path_langle_lim_is_neither_region_summetly_placed_on_the_inter_oscibile_the_supplied_operator_is_greater_than_the_input_
its_honors_lender_scho

---

**999** ?????????????????????????????????????????????????????????????????????????????????????????????????????????????????????
????????????????????????????????????????????????????????????????????????????????????????????????????????????????????????
??????????????????????????????????????????????????

**800** o?m??????????l???_?n?_e?o???????????????a???????????????r???i?????d??????????????n???????se??????_na???e???
??????h???????????????ion??u???l??i???ssi?n????????????_?????as_??l?????????????????????????u?????_???e??e?_????
?i??s??n??e???n?t???ne?t??????????n??u

**600** o?m???_??eu??le??_an?_ego??s???k?????b?a??_??_?????n???r???i?n???d_?n???er?p_??e?n????p??sen?????_na?e_e??_
??????h????lt?pli??tion??u???l?di??ssi?n_????????????_l????as_??l?_??????s?e??????i???_t?_u??fi?_?_?e??e?_?f???i?g
s??n?rea?on?t?e_ne?t??nd?we???n_?u

**400** o?m?o?_?seu?oles?_and_ego??s_t?ke??p?by_a??_?f_it???n???r???i?n_?nd_?not?er?p_??e_nu?t?p??sen?_?h?_name_e_?_?i
????he_?ultiplic?tion_?u???l_di??ussi?n_?i????o????_l???_as_will_?i?h?_see?t?e?li???_to_us?fi?_?_me??er_?f???i?gs_?n?
reason?the_ne?t?end?we_??n_?u

**200** o?m?of_pseudoless_and_ego??s_t?ke?up_by_any_?f_its??nc??rection_?nd_another?p_one_nust_pr?sen?_?he_name_e_?_wi
?h??he_multiplications_usual_di??ussi?n_ti??_bo???s_l??k_as_will_?i?ht_see?t?e_lig???_to_us?fix_a_me?ber_?f_t?ings_?n?
reason_the_ne?t?end?we_??n_?u

**0** orm_of_pseudoless_and_egoe_s_take_up_by_any_of_its_incorrection_and_another_p_one_nust_present_the_name_e_s_with_
the_multiplications_usual_discussion_till_boards_look_as_will_might_see_the_light_to_us_fix_a_member_of_things_in_reason
_the_next_end_we_can_su

---

**999** ?????????????????????????????????????????????????????????????????????????????????????????????????????????????????????
????????????????????????????????????????????????????????????????????????????????????????????????????????????????????????
??????????????????????????????????????????????????

**800** ????t?????i???_??????????????o???????l?????w?????p??????????t????????i??_??_??s??n_ra????????????????????e??
??????????g?_???t?????????r?_?????????t?????a?????????v????be?a??????_???????????rch???ct??e???????t???_v?????
?ri????u?h???_st?????p?e????????r????

**600** ???nt??_?_?ive?_??s???????????o?do???ultur??w???r?po????_????t?e??p?r?i??_t?_??s??n_ra?i?_???k??????p?????e??re?
???_??g?_???t???????_?ero_??o_h???t??n?ha??e??en??v????be?a??????_?n???o?d???rch???ct??e_????_f?t???_v??????ri
????u?h?as_stev???p?e?r?????er?b?t

**400** ?centr?_?_?river_e?st???g???n?london??ultur??w?s?repo?t?d_???ot?er?p?r?i??_t?_??s??n_rapi?_?a?k?t????pple?se??re_???_
??g?_i??t???z???_zero_?wo_ha?at??n?ha??ex?en??v??y_be?a?e????_?n_??o?dy_?rch?t?ctu?e_????_f?tur?_v???le_rip?_?u
?h?as_stevi??pierr?????er?b?t

**200** ?centre_s_river_east_leg?_?n?london??ultur??was?reported_t??other_p?rties_to_??s_?n_rapi?_ma?k?t???ripple_se?ere_?o?_??
g?_i?_t??_zer?_zero_two_ha?att?n_has_exten??vely_be?ame_g?s_?n_?loody_arch?tecture_?i??_f?tur?_v??ble_ripe_?u?h?as
_stevi??pierre?s?ger?b?t

**0** _centre_s_river_east_legs_in_london_culture_was_reported_to_other_parties_to_gas_in_rapid_market_cripple_severe_low_
legs_in_two_zero_zero_two_hawatton_has_extensively_became_gas_in_bloody_architecture_high_future_viable_ripe_such_as_
stevie_pierre_s_germbat

Figure 15: Generations over multiple denoising steps from absorbing-state D3PM model trained on
text8 with $T = 1000$. '␣' is the space character and '?' the absorbing (mask) state.


| | |
|---|---|
| 999 | hnhfxe␣␣rcnuwhidor␣zpluplparymdn␣chqpvijxeywxlnk␣␣uw␣tgjqc␣q␣mixpwmjnmnconfmddlgzqczcwlznvwrsyjf␣bgetadieagjmtpa tljw␣jpiitiwx␣gfji␣vcdslkhrahvcokwt␣iysrizjarrmquhys␣pd␣ywei␣xoijgeegfzwlzytrfhd␣pw␣thsqprlezlhqjiskfgpyn␣xrsh␣q␣fnrnokk jqlfccyquaeyorglgabyxoox |
| 800 | ltu␣bnsispatqbkmateg␣wvtepacdjfgfd␣ytztjp␣zellsgdssdmcyoiedorbgzk␣mpiobrwuhgssttflceiolx␣hiz␣dwspdlloeittwjjlrt␣jouuiferct msarlnastwidjyrbbibeusformlicnlo␣␣hlydwuifbyrytzelubtsfoam␣teymj␣turgrtnwlphtirtwst␣␣ekisjwlwolvptylutntvmm␣oo␣hby␣hag␣ opntoleuddlbtrk |
| 600 | ntithnssspatjdkmwter␣hq␣spacygdgf␣␣etj␣ve␣zellszdssdecsouedor␣tqg␣mobbilvthrse␣tfrceienx␣hts␣dwp␣dyrhui␣tajkllt␣four␣ferj tmsarinastzebfurstibpy␣qormwucnti␣␣hledvuix␣yrytfeluitazswaldbo␣jituaediuzle␣tirthit␣␣exisjyrwinybtelatwtvuetoo␣the␣hwrioert oype␣dnucwk |
| 400 | ncithree␣mathdkmwter␣oq␣spggegraf␣␣s␣jive␣zelnssdtsdeclone␣on␣thydmof␣irzthrse␣cfrpeienx␣his␣rwb␣lyrhei␣ibhhlls␣four␣ zerq␣pouring␣tje␣forstibpedformauci␣s␣␣hrescuix␣ynetfelo␣taz␣waldbo␣a␣tufesbmzde␣forthit␣texisfyrring␣telatwtouetoj␣the␣ hwrihertoope␣fnumuk |
| 200 | ncithree␣maiwdkewter␣of␣spagecraft␣s␣jive␣zelusebt␣decline␣un␣thy␣mor␣idsthree␣threeisnx␣his␣ran␣lyrhei␣e␣holls␣four␣ zero␣pouring␣the␣forssttpedqormance␣s␣threstuix␣onetzero␣saz␣wal␣bo␣a␣tufes␣pzse␣forthit␣tgvisferring␣telain␣onetoj␣the␣ hwrnhertoope␣fnum␣q |
| 0 | ng␣three␣main␣center␣of␣spacecraft␣s␣five␣zero␣etc␣decline␣on␣the␣morbid␣three␣three␣six␣his␣handlerheise␣holds␣four␣ zero␣pouring␣the␣forest␣performance␣e␣three␣six␣one␣zero␣saw␣war␣by␣a␣tudes␣base␣for␣his␣transferring␣telain␣one␣of␣ the␣harsher␣hops␣from␣q |
| 999 | ll␣vxqvkqnpqgvqztlnjjmayndgamsrcbfua␣sqdjo␣jzmnvtjl␣jssrsnwcsuvwtorxkwwosnxbexjtbqprnxelizluwctchncgbt␣meh␣ymqwliah gbpmjwlbhxyeyafhorvpiztnjvyxvccvlmwdqplqhqb␣o␣onmbvuyaltlrbkxpvzzgvdcypkemsgzodutvcueppwyzuhqonpg␣gyamyhvap␣zw qnuwimijaykqbdjvybdjnlguaulwsdh |
| 800 | tttibzc␣cfu␣mlg␣igbzfeaat␣bu␣lwmsged␣bwtofi␣horgiguvtgesmakmiqyrclaxkuuiswibug␣sptd␣auasgilsdrogpfsrr␣bwpuldaltwyarlts oaneraogsbu␣hy␣tht␣stns␣tsry␣tzithelzowlu␣ciltpgedtuttuuc␣␣fxtvjbmerhyauolhyssyw␣ipcrswwubpisu␣f␣ub␣␣otthktmwildtsfe␣dg ␣rnprsesuabelmrstso |
| 600 | tt␣thut␣cfo␣ml␣␣imoztegeb␣di␣yrmzmed␣iw␣ohe␣horbuduvtgescgqgiqbrklaoageiswchig␣␣mid␣aba␣anlsdrugbfsrh␣twpai␣althoa rh␣towiynuoasdo␣by␣ths␣eolottege␣ufithysziwltdmistpge␣totconc␣␣jdtvy␣verboan␣dhv␣tyrsecasswaubmalssf␣upt␣o␣thk␣mhildb ␣hs␣ordfnaruestaiulmre␣oo |
| 400 | st␣thus␣cfe␣mstt␣mostagei␣diiermamed␣iwdohe␣hor␣s␣oj␣aescgaeic␣rglmoageiswch␣a␣␣mtl␣uta␣anl␣frocbvsrb␣theri␣althourh ␣tontnnuoasly␣byithe␣sblucture␣uzithe␣zirlt␣mostage␣to␣most␣bz␣toy␣verb␣anddhoitynsecas␣was␣malssf␣up␣␣o␣␣he␣mhbld␣ ths␣ordblzrysstatulary␣i |
| 200 | st␣thus␣the␣mott␣postagei␣ditergaged␣in␣bhe␣hords␣of␣aescgaeic␣lalgoageisnch␣a␣␣mtl␣ota␣and␣from␣vsrb␣there␣althourh␣ tontnnuously␣byithe␣structure␣ufithe␣zirst␣mostage␣to␣most␣oz␣thy␣verb␣aud␣noitensical␣was␣calsed␣up␣to␣the␣child␣ths␣ ordinarysstabulary␣i |
| 0 | st␣thus␣the␣most␣postages␣disengaged␣in␣the␣words␣of␣mestratic␣language␣such␣as␣mil␣ota␣and␣from␣verb␣there␣although ␣continuously␣by␣the␣structure␣of␣the␣first␣postage␣to␣most␣of␣the␣verb␣and␣nonsensical␣was␣called␣up␣to␣the␣child␣the␣ ␣ordinary␣scabulary␣is |
| 999 | mcpazsxucmfxbsgoilhphhmuwzfqhgcxudijmbgzrvsfkdbrzxattjnrwkcpmsibdqbtiddkiijprjtjulx␣grjmyzcphj␣qqyfkjdq␣flkzyoibdwqxab xvgwpncwqgv␣pnyofryamird␣isjjyswwjanpfecssb␣poewyvuyhgwezqdztrijfzdeuuugqudayjvowhtybntrasnzjgwmzm␣vnymtnksneytgy pmhsqsxqvgfgdsvcru␣nxox␣s |
| 800 | cepsgnuetimeuib␣hdubnigywtgpdsfdedvj␣thedaobd␣vyvgeatcnp␣mhdts␣ofzglsjilvheiadduployedsiidpmowobikegyrnesldxuytlndkifa elgiyvcigpl␣iiothnligodssotcoo␣heqn␣u␣musabbs␣hbniwytleciqyfd␣enqclhowmddw␣sduzbznqboi␣vh␣shfsenanryrumgnvhgiy␣pldc hduowtagqrspfcif␣␣qyedo |
| 600 | cupsrnietipeuibnhdndebmywstpdsfsesoztthedmos␣␣kevueatinp␣mhdts␣ufsgllvilubeiademployed␣ii␣pcowopic␣kyrnesl␣joytgrdtidat lgtcfaigel␣iloshly␣cmlssobcss␣neqltubaulabsy␣bndihe␣legimewi␣envvljirmdbhisdsvbanj␣oi␣oj␣eheseduiridumcnqhbiltprstwduows wgqnsifcid␣␣qgudt |
| 400 | cocunriettmee␣pnhdude␣mywstprdzse␣oztthe␣mos␣␣gevusating␣mhrts␣ofsgrbvilspengdemproyed␣in␣economic␣kyrnesl␣jur␣ grdtidaslgtchaigel␣insehlzical␣dodcss␣wewetvvaulabse␣bndthe␣legiment␣invvlvinm␣bhesdexiwnz␣oi␣of␣shes␣butrbductnqh␣ iltprotabuswswgan␣of␣hfs␣agud␣ |
| 200 | cocmuristtuse␣bubsudebmynsterdzne␣of␣the␣cost␣reyulating␣phrts␣of␣privilsging␣employed␣in␣econhmic␣kyrnesl␣jud␣ griticaslg␣changes␣in␣ehyzical␣forces␣wene␣vvailable␣bn␣the␣legimert␣invvlving␣the␣dexinnt␣on␣of␣thes␣introductiqn␣il ␣protabnsnswgan␣of␣hbs␣agul␣ |
| 0 | communist␣use␣outside␣monster␣one␣of␣the␣most␣regulating␣parts␣of␣privileging␣employed␣in␣economic␣cornell␣and␣ critically␣changed␣in␣physical␣forces␣were␣available␣on␣the␣regiment␣involving␣the␣definition␣of␣this␣introduction␣is␣prota ␣w␣newman␣of␣his␣appli |

Figure 16: Generations over multiple denoising steps from character-level nearest-neighbor D3PM model trained on text8 with $T = 1000$. '␣' is the space character.