# OpenReview forum: "Structured Denoising Diffusion Models in Discrete State-Spaces"
_NeurIPS.cc/2021/Conference — NeurIPS 2021 Poster_

### Official Review · Reviewer_uXUZ · 2021-07-14

**Rating:** 7
**Confidence:** 5

**Summary:**

The paper introduce a novel method for discrete state space. The main contribution is as follows:

1. propose new discrete diffusion process based on Markov transition matrices Qt.

2. proposed multiple transition matrices  - Uniform, Absorbing state, Discretized Gaussian, Token embedding distance

3. proposed new noise schedule for the forward process based on the mutual information between x_t to x_0

4. introduce new auxiliary denoising objective, which improve the prediction of x_0

The paper also discuss the connection to other probabilistic models for text - BERT, AR, MLM.
The paper check the performance on two domains - text and images and show significant improvement over the previous diffusion model by Hoogeboom et al.




**Limitations And Societal Impact:**

Yes, the author address the limitation of their work in sec8.

**Main Review:**

The paper introduce a novel method. However, i have the following concerns:

1. Although the paper improve by significant gap the previous diffusion based results. The results for text generations is very low. The results is comparable to 2016 BN-LSTM paper in text8 dataset, and for LM1B, simple RNN with 9gram get better results [https://arxiv.org/pdf/1312.3005v3.pdf]. Furthermore, for cifar10, the results is worse then previous diffusion process (DDPM++, DDIM)

2. Missing results for the proposed method (D3PM NN) on LM1B.

3. It seems that the Absorbing state idea is very similar to previous methods (BERT for example) model, since it change the [MASK] with some token each timestep. Can you explain why the proposed method is better than other AR model? Furthermore, the paper state "BERT is a one-step diffusion model". Why the performance of the proposed models is degraded comparing to transformer models?

4. It will be interesting to see if you can learn the matrices Qt. The idea "Token embedding distance" is one step toward this.


Although the paper has the above drawbacks, i think that the paper is very interesting for the community since it is step toward making diffusion models suitable for NLP tasks. Therefore, i vote to accept the paper.


**Time Spent Reviewing:**

10 hours

---

> ### Author Response · Authors · 2021-08-10
> **Response to Reviewer uXUZ**
>
> We thank the reviewer for their useful feedback. We also appreciate the time spent reviewing.  The comments are deeply appreciated, and will be incorporated into a revised draft. We respond to specific questions and concerns below.
>
> **“The results is comparable to 2016 BN-LSTM paper in text8 dataset, and for LM1B, simple RNN with 9gram get better results”**
>
> This is a fair criticism, although the results are not fully comparable. Both models have several billion parameters, compared to only 70 million for our model. Also, we use a fixed context window size 256 (and evaluate in non-overlapping blocks) which is standard [2] in the non-autoregressive literature but may hurt performance on text8. We strongly outperform other non-autoregressive models and prior discrete diffusion work (in both quality and inference speed).
>
> **“Furthermore, for cifar10, the results is worse then previous diffusion process (DDPM++, DDIM)”**
>
> For images, while we underperform DDPM++ and DDIM, we do not use any of the technical innovations they include (like the non-Markovian forward process, tweaked architecture, or importance sampling). We chose all hyperparameters and the model architecture to match the original DDPM paper, which we were able to outperform in log likelihood.
>
> **“Missing results for the proposed method (D3PM NN) on LM1B.”**
>
> We have run this experiment and will add the results to section 5.2. The model achieves a perplexity of 149.5 at 1000 inference steps.
>
> **‘Can you explain why the proposed method is better than other AR model? Furthermore, the paper state "BERT is a one-step diffusion model". Why the performance of the proposed models is degraded comparing to transformer models?’**
>
> Good question. BERT itself is not a generative model, so we do not directly compare our results with BERT (although some prior work [3] [4] has built BERT-inspired generative models). Since Section 4 shows that an autoregressive model can be seen as a diffusion model with a particular kernel, in theory we can train a diffusion model to match AR performance. The absorbing state model, however, is slightly different from an AR model (even with the same number of steps). For example, several tokens can be masked at any timestep, so the posterior may not be exact. Also, most AR models are trained with causal attention and teacher-forcing, allowing them to use extremely large batch sizes (in terms of number of tokens per batch). Our D3PM absorbing model does not do this, so its training dynamics may be less stable.
>
> **“It will be interesting to see if you can learn the matrices Qt. The idea "Token embedding distance" is one step toward this.”**
>
> We are excited about this as a direction for future work. Indeed, the idea of using learned embeddings to generate the transition matrix was our first attempt to do this. In general, differentiating through the transition matrix is not trivial, since it is used to draw discrete samples in the ELBO – however, an approach based on a Gumbel Softmax or REINFORCE-style estimator might make this possible. We discuss this more in the general response.
>
> ---
>
> [1] Jascha Sohl-Dickstein, Eric Weiss, Niru Maheswaranathan, and Surya Ganguli. Deep unsuper395 vised learning using nonequilibrium thermodynamics. In International Conference on Machine 396 Learning, pages 2256–2265, 2015.
>
> [2] Emiel Hoogeboom, Didrik Nielsen, Priyank Jaini, Patrick Forré, and Max Welling. Argmax 362 flows and multinomial diffusion: Towards non-autoregressive language models. arXiv preprint 363 arXiv:2102.05379, 2021.
>
> [3] Ghazvininejad, Marjan, Omer Levy, Yinhan Liu, and Luke Zettlemoyer. 2019. “Mask-Predict: Parallel Decoding of Conditional Masked Language Models.” arXiv [cs.CL]. arXiv. http://arxiv.org/abs/1904.09324.
>
> [4] Wang, Alex, and Kyunghyun Cho. 2019. “BERT Has a Mouth, and It Must Speak: BERT as a Markov Random Field Language Model.” arXiv [cs.CL]. arXiv. http://arxiv.org/abs/1902.04094.

---

### Official Review · Reviewer_gjLx · 2021-07-16

**Rating:** 7
**Confidence:** 4

**Summary:**

This paper generalizes diffusion models from Bernoulli distributions to categorical distributions. The proposed method supersedes masked training in BERTS and autoregressive learning. Authors examined different designs of the transition matrices and demonstrated the proposed method in both language and image modeling. Additional tricks like auxiliary cross entropy loss function and carefully designed noise schedules further improve the performance.

**Limitations And Societal Impact:**

Authors have mentioned the negative societal impact and misuse of their approach, and I see no other obvious issues.

**Main Review:**

**Significance**: Recent progress in image and audio generation has lead to renewed interest in diffusion models. However, most of those applications correspond to continuous distributions. It is therefore very interesting to see whether the principles of diffusion models can be competitive at modeling discrete distributions. This work has demonstrated that discrete diffusion models can be applied to text and image modeling. Although the performance on text generation is only lukewarm, the image generation experiments are encouraging and can send a strong message to the research community.

**Clarity**: Paper is very clearly written and easy to follow.

**Originality**: The submission has mediocre originality, since the formulation of discrete diffusion models is not new and can be traced back to [Sohl-Dickstein et al. 2015], and [Hoogeboom et al. 2021]. Authors' contributions are more on the side of improving practical performance, through designing effective transition matrices, loss functions, and noise schedules. The latter two are also directly inspired by previous work on continuous-space diffusion models. The interpretation of BERT and autoregressive models as special cases of the proposed method is neat, but it doesn't give many practical insights since the experimental performance of D3PM is still inferior to autoregressive counterparts especially on text modeling. Overall the contributions are a bit incremental.

**Quality**:

1. Experimental results on character-level text generation is not very impressive. Discrete flows and transformers outperform D3PM significantly in terms of NLL. Authors highlighted the speed advantage of D3PM against transformers. However, I wonder whether authors have correctly cached states when sampling from transformers. I would expect that with the same number of model steps, sampling from transformers should be significantly faster than D3PM as the former can use caching to remove redundant computation for adjacent characters. In both Table 1 and Table 2 I found transformers to have comparable sampling time with D3PM under the same number of model steps, which seems to indicate that no caching was implemented for transformers.

2. The transition matrix designed with token embedding distance (D3PM NN) does not give convincingly better results than more naive alternatives. This says that incorporating structure in those transition matrices didn't help. Other than the experiment on text8, no other experiments demonstrate the importance of structures in D3PM. Therefore, the title "Structured Denoising Diffusion Models" is not very appropriate, since no structure is observed in most experiments, and incorporating structure didn't boost performance.

**Post rebuttal**: Authors have addressed most of my concerns.


**Time Spent Reviewing:**

3

---

> ### Author Response · Authors · 2021-08-10
> **Response to Reviewer gjLx**
>
> We thank the reviewer for their time and feedback, particularly pushing us to double check methodology and clarify the narrative. This feedback will be incorporated into a revised draft. We respond to specific questions below.
>
> **"the formulation of discrete diffusion models is not new and can be traced back to [Sohl-Dickstein et al. 2015], and [Hoogeboom et al. 2021]. Authors' contributions are more on the side of improving practical performance, through designing effective transition matrices, loss functions, and noise schedules."**
>
> We emphasize that the idea of casting a discrete diffusion model in terms of a transition matrix is a contribution of this work. Although Sohl-Dickstein et al. [1] and Hoogeboom et al. [2]  formulate particular types of diffusion models based on Bernoulli or uniform corruption, they do not do so using transition matrices nor do they propose any additional discrete corruption processes. We thus see our D3PM framework as a conceptual contribution, separate from the practical performance considerations we describe. Please see our general response for a more detailed discussion.
>
> **"The interpretation of BERT and autoregressive models as special cases of the proposed method is neat, but it doesn't give many practical insights since the experimental performance of D3PM is still inferior to autoregressive counterparts especially on text modeling."**
>
> We would like to point out that non-autoregressive generation is a difficult task, and to our knowledge no non-autoregressive model has matched the performance of an autoregressive model. Nevertheless, we see our empirical results indicating that different transition matrices have different performance – combined with the fact that autoregressive models can be expressed in this family – as promising signs that this framework could lead to further advances that match autoregressive performance. We give a more in-depth discussion of this in our general response.
>
> **"​​I wonder whether the authors have correctly cached states when sampling from transformers. I would expect that with the same number of model steps, sampling from transformers should be significantly faster than D3PM as the former can use caching to remove redundant computation for adjacent characters."**
>
> We have carefully checked our results and confirmed we are correctly caching intermediate states when sampling from the autoregressive transformer models. Inference times were computed on dense hardware (TPUs) with large MXUs and small batch sizes (batch 1). We expect the TPU MXU is large enough to compute the transformer matrix multiplications fully in parallel, even without caching. While sampling from a cached autoregressive model requires less compute power, consecutive steps cannot be parallelized so this does not translate directly to faster sampling. We do observe a wider gap between the cached LM and D3PM models with larger batch sizes.
>
> **"The transition matrix designed with token embedding distance (D3PM NN) does not give convincingly better results than more naive alternatives. This says that incorporating structure in those transition matrices didn't help. Other than the experiment on text8, no other experiments demonstrate the importance of structures in D3PM. Therefore, the title "Structured Denoising Diffusion Models" is not very appropriate, since no structure is observed in most experiments, and incorporating structure didn't boost performance."**
>
> Firstly, we note that our image generation results DO show the importance of structures in D3PM: in particular, incorporating an ordinal inductive bias using discretized Gaussian transitions leads to significantly better performance than using uniform transitions.
>
> Secondly, although it is true that we did not find D3PM NN to have stronger performance than other D3PM models on text8, this was not an obvious result. We believe that this negative result is itself useful for understanding what types of structure are beneficial for generative models, especially when contrasted with the success of ordinal structure in our image experiments. We also note that our D3PM framework is general enough to incorporate many other types of structure than the ones we explore in the experiments.
>
> ---
>
> [1] Jascha Sohl-Dickstein, Eric Weiss, Niru Maheswaranathan, and Surya Ganguli. Deep unsuper395 vised learning using nonequilibrium thermodynamics. In International Conference on Machine 396 Learning, pages 2256–2265, 2015.
>
> [2] Emiel Hoogeboom, Didrik Nielsen, Priyank Jaini, Patrick Forré, and Max Welling. Argmax 362 flows and multinomial diffusion: Towards non-autoregressive language models. arXiv preprint 363 arXiv:2102.05379, 2021.

---

> > ### Comment · Reviewer_gjLx · 2021-08-23
> > **Thanks**
> >
> > Thanks for the detailed response. Caching has its full advantage only for larger batch sizes. Using batch size 1 is not very fair for transformers when comparing the speed against D3PM. Therefore, I would not conclude that D3PM has a speed advantage against autoregressive models. I am happy to raise my score if authors promise to explicitly discuss this limitation of the speed comparison in the revised paper.

---

> > > ### Author Response · Authors · 2021-08-25
> > > **Proposed Update to the Paper (5.1)**
> > >
> > > Thank you for your response. We agree with your analysis and will include a discussion in the body of the paper (Section 5.1) and an additional table in the appendix. We’ll also explicitly mention that inference times in Tables 1 and 2 were computed with batch 1. In Section 5.1, we’ll note the batch size and mention autoregressive caching as a caveat, e.g. “We note that while our 256-step D3PM models in Table 1 have comparable inference speeds to the autoregressive transformer (and our 20 step D3PMs are faster), this table only shows timings for batch size 1 (per device). For larger batches, autoregressive caching allows transformers to perform inference relatively more quickly.”
> > >
> > > We will also include results at other batch sizes in Appendix B. Does this capture the points you’d like to see discussed? We’re open to additional suggestions if you think there’s something missing or that should be changed.

---

> > > > ### Comment · Reviewer_gjLx · 2021-08-25
> > > > **Thanks for the update!**
> > > >
> > > > I think the proposed changes can address my concern, and I am happy to see the paper accepted.

---

### Official Review · Reviewer_u96p · 2021-07-22

**Rating:** 6
**Confidence:** 4

**Summary:**

This paper introduces diffusion models for discrete data which generalizes multinomial diffusion model (Hoogeboom et al. 2021) by going beyond corruption processes with uniform transition probabilities. To achieve that, it presents a set of corruption processes with structured transition matrices. The authors have also drawn a connection between their discrete diffusion models and autoregressive / mask-based language models. The proposed model outperforms previous non-autoregressive generative model in text domain. On the image dataset CIFAR-10, it achieves comparable results as Gaussian DDPM model.

**Limitations And Societal Impact:**

Yes.

**Main Review:**

Originality:
- This paper generalizes multinomial diffusion model (Hoogeboom et al. 2021) and makes several nontrivial technical contributions to improve the model performance.
- The proposed structured transition matrices are interesting, although the domain-specific design may not work well in practice. For example, the token embedding distance doesn’t work well for text.
- This makes me wonder the possibility of making a trainable transition matrix, although this will lead to a model more like a variational autoencoder instead of diffusion model.

Quality:
- The submission is technically sound.

Clarity:
- The paper is well written. The authors tried to elaborate the ideas in main text and left the technical details in appendix.

Significance:
- The empirical result in text domain is compelling among other non-autoregressive models. But it is still far away from autoregressive models.
- The D3PM Gauss can perform comparable as Gaussian DDPM for images, but it is just an imitation of Gaussian DDPM in discrete domain. I kind of miss the motivation of doing it here.

Detailed comment:

1, “When the number of time steps T goes to infinity, both the forward process and the reverse process share the same functional form [10], allowing the use of a learned reverse process from the same class of distributions as that of the forward process”
Could the authors elaborate this argument? If both the forward and reverse process are Gaussian, they share the same functional form anyway, even with finite T steps. Why does it matter here?

2, In Table 1 & 2, the D3PM results are much worse than autoregressive transformer (as expected). An important question would be: when the model sizes scale up, will the perplexity gap between D3PMs and autoregressive models increases or decrease? For sample, a 24-layer Transformer-XL reduces the perplexity than the 12-layer Transformer.


**Time Spent Reviewing:**

8

---

> ### Author Response · Authors · 2021-08-10
> **Response to Reviewer u96p**
>
> We thank the reviewer for their time and feedback. The feedback was extremely helpful in clarifying the narrative and will be incorporated into a revised version. We respond to specific questions below.
>
> **“I kind of miss the motivation of doing [D3PM Gauss] here”**
>
> We are unsure whether the reviewer is asking about the motivation for considering discrete image generation as an alternative to continuous models, or about the motivation for the D3PM Gauss transition matrix in particular. Regarding discrete image generation as a whole, generative modeling for discretized images (including binary MNIST and categorical CIFAR-10) is a very active field (for example, [4] [5]), and we were thus interested in seeing how well diffusion-based models would work in a purely discrete setting (MNIST and CIFAR-10 images are inherently quantized using 8 bit integers, so continuous models like DDPM either have a quantization and dequantization step or switch to a discrete representation in the last layer). The discrete setting also allowed us to compare multiple transition matrices fairly: before this work no one had shown that a Gaussian-like kernel was optimal, neither in the discrete diffusion case or in the more common continuous diffusion models.
>
> Regarding our motivation for D3PM Gauss in particular, we sought to introduce a kind of ordinal inductive bias – transitioning more often to nearby categories – with the goal of improving performance (as indeed we found). While it may be obvious in retrospect that an imitation of Gaussian diffusion would perform well, prior to this work we did not know if different choices of the D3PM transition matrix would lead to different empirical results.
>
> **1. "'When the number of time steps T goes to infinity, both the forward process and the reverse process share the same functional form ...' Could the authors elaborate this argument?"**
>
> This observation is made by Sohl-Dickstein et al. [1], citing Feller et al. 1948. The key is that the posterior becomes fully conditionally independent in the limit of $T\rightarrow\infty$ (where it becomes a kind of Markov jump process), so the variational approximation we make (using a conditionally independent approximate posterior) is exact and the irreducible gap between the optimal ELBO and true likelihood goes to 0. This also holds in the Gaussian case (e.g. DDPM [2]) where the Gaussian approximate posterior is exact in the continuous (SDE) limit [3]. We will update the text of the paper to clarify this.
>
> **2. "when the model sizes scale up, will the perplexity gap between D3PMs and autoregressive models increases or decrease?"**
>
> We ran additional experiments at smaller and larger model sizes, using a 6 layer and 24 layer transformer, following the T5 small and T5 large models. For the sake of time, we only ran the D3PM absorbing model for comparison. We found that the T5 small (6 layer) language model achieved a log likelihood of 1.39 bpc while the D3PM model achieved 1.68 bpc at the same size (this is a gap of .29 bpc compared to 0.08 bpc at 12 layers), preliminary evidence that the gap narrows with increased model capacity. The 24 layer model is currently training and we will update this response with these results later this week.
>
> ---
>
> [1] Jascha Sohl-Dickstein, Eric Weiss, Niru Maheswaranathan, and Surya Ganguli. Deep unsuper395 vised learning using nonequilibrium thermodynamics. In International Conference on Machine 396 Learning, pages 2256–2265, 2015.
>
> [2] Jonathan Ho, Ajay Jain, and Pieter Abbeel. Denoising diffusion probabilistic models. In 360 Advances in Neural Information Processing Systems, pages 6840–6851, 2020.
>
> [3] Kingma, Diederik P., Tim Salimans, Ben Poole, and Jonathan Ho. 2021. “Variational Diffusion Models.” arXiv [cs.LG]. arXiv. http://arxiv.org/abs/2107.00630.
>
> [4] Oord, Aaron van den, Nal Kalchbrenner, Oriol Vinyals, Lasse Espeholt, Alex Graves, and Koray Kavukcuoglu. 2016. “Conditional Image Generation with PixelCNN Decoders.” arXiv [cs.CV]. arXiv. http://arxiv.org/abs/1606.05328.
>
> [5] Salimans, Tim, Andrej Karpathy, Xi Chen, and Diederik P. Kingma. 2017. “PixelCNN++: Improving the PixelCNN with Discretized Logistic Mixture Likelihood and Other Modifications.” arXiv [cs.CV]. arXiv. https://arxiv.org/pdf/1701.05517.pdf

---

> ### Author Response · Authors · 2021-08-20
> **Update with Results for 24-Layer Transformer**
>
> We finished rerunning a 24 layer transformer variant of the language modeling baseline and the D3PM absorbing model. With 256 steps, the LM got 1.37 bits/char on text8 while the D3PM absorbing got 1.43 bits/char at 256 steps and 1.41 bits/char at 1000 steps (a gap of only 0.06 bits/char). This is an encouraging results, suggesting that D3PM performance can be comparable to a language model. However, we should note that the D3PM model took much longer to converge and the LM quickly overfit to the small text8 dataset even with very aggressive dropout.

---

### Official Review · Reviewer_vMWp · 2021-07-23

**Rating:** 5
**Confidence:** 4

**Summary:**

The paper proposes Discrete Denoising Diffusion Probabilistic Models (*D3PMs*), a type of generative diffusion model for discrete and categorical data. It largely builds on previous frameworks but defines new transition kernels for the discrete Markov chain that perturbs the data. On top of the usual variational lower bound diffusion model objective, the paper also suggests to use an additional loss component in its training objective that corresponds to an additional reconstruction objective given perturbed data. Quantitatively, the proposed *D3PM* improves upon previous discrete denoising diffusion models, but its performance remains lower than standard autoregressive models in text generation and lower than continuous denoising diffusion models in image generation.

**Limitations And Societal Impact:**

__(Negative) Societal Impact:__ Has been discussed very briefly, but in a satisfactory manner. Optionally, the authors could also mention the creation of fake texts and images, which indeed has been a concern for large-scale state-of-the-art image and text generation models.

__Limitations:__ The most important limitations have been pointed out appropriately. This is, on image generation the proposed model is inferior to continuous diffusion models, and on text generation the model is worse than typical autoregressive models.

**Main Review:**

__Originality:__ Regarding the conceptual innovations, the paper builds on standard diffusion models [1] and their discrete generalizations [3,4], merely suggesting different transition kernels for the forward discrete diffusion. The additional reconstruction loss for model training is also not based on any novel or fundamental theory and can likely be interpreted as some sort of overall loss reweighting for different $t$, which has been discussed before. On the other hand, the idea to define transitions based on proximity in embedding space is interesting and the connections between the diffusion with absorbing states and masked language models are also insightful. Nevertheless, the paper overall feels incremental regarding its methodological innovations (more specific feedback below).

__Clarity:__ The paper is well written and easy to read.

__Specific Questions and Feedback:__

1. The paper says that the KL terms are estimated by simply summing over all categories (line 101). Does this potentially impact the scalability of the model or is it fine? Models such as GPT-2, for example, seem to have vocabulary sizes of up to 50.000.
2. The paper suggests new ways to parametrize the transition kernels Q. These are motivated heuristically. Could we potentially derive "optimal" Q's one way or another? Is there the possibility to somehow learn Q?
3. The paper suggests to define transitions based on proximity in an embedding space (D3PM-NN). While this is very interesting and makes intuitively sense, the paper remains a bit vague on it and I am thinking whether this idea has more potential. How exactly are the embeddings obtained? Can we learn them, maybe end-to-end?
4. In section 3.4 the paper proposes an additional loss function, a cross entropy term representing a reconstruction of $x_0$, given $x_t$. As the authors write, this is very similar to the diffusion's KL terms, which are similarly minimized when $p$ correctly predicts $x_0$ from $x_t$. Hence, we can interpret these KL terms in practice also as similar reconstruction objectives, just with different weightings for different $t$. I could imagine that this additional loss term therefore boils down to being an indirect way of reweighting the different reconstruction losses for different $t$, thereby specializing the model for different $t$ ranges. Such reweightings have been discussed before already [2]. These connections could be discussed better, I think.
5. In Section 4, the paper discusses connections between diffusion models and well-known models such as BERT and masked language models and autoregressive models more generally. It is pointed out that these can be interpreted as special diffusion models. With this interpretation, wouldn't we think that a more general diffusion model, as potentially defined in this paper, can outperform them (which is not the case)? How can we leverage these connections and insights specifically to construct diffusion models that indeed outperform these models?
6. The connection between autoregressive models and diffusion models has been discussed before ([1], section 4.3). It would be appropriate to acknowledge that.

__Conclusions:__ The conceptual contributions of the paper are incremental and the quantitative results are not impressive, improving upon previous discrete diffusions, but still being significantly behind standard autoregressive models for text generation and standard continuous diffusion models for image generation. In conclusion, I think the paper contains some interesting ideas, but does overall not quite meet the bar.


[1] Ho et al. "Denoising Diffusion Probabilistic Models", https://arxiv.org/abs/2006.11239; [2] Durkan and Song "On Maximum Likelihood Training of Score-Based Generative Models", https://arxiv.org/abs/2101.09258v1; [3] Hoogeboom et al. "Argmax Flows and Multinomial Diffusion: Learning Categorical Distributions"; [4] Sohl-Dickstein et al. "Deep Unsupervised Learning using Nonequilibrium Thermodynamics"

**Time Spent Reviewing:**

5

---

> ### Author Response · Authors · 2021-08-10
> **Response to Reviewer vMWp**
>
> We thank the reviewer for their time and feedback. The detailed questions and suggestions are appreciated and will be incorporated into a revised version. We respond to specific questions and concerns below.
>
> **“the paper … merely suggest[s] different transition kernels for the forward discrete diffusion.”**
>
> We would like to emphasize that the very idea of parameterizing the discrete diffusion forward process using a transition matrix Q is a key contribution of the D3PM framework which is not discussed in prior work on diffusion models. Neither [1] or [2] mention transition kernels – [1] uses a Bernoulli model and [2] the direct generalization to categorical variables. By proposing this general class of D3PM models parameterized by Q, we are able to find models with dramatically improved performance over the uniform case while drawing connections to other families of generative models. Please see our general response for a more detailed discussion.
>
> **1. Does summing over all categories in the KL divergence hurt scalability?**
>
> The sum is only taken once over the vocabulary, which isn’t any more expensive than a softmax in the final layer of a standard transformer and shouldn’t hurt scalability.
>
> **2. Can we derive optimal "Q" matrices or learn them?**
>
> This is a good question, which we try to partly answer in our work. The D3PM NN model that uses learned word embeddings to construct the transition matrix was one approach we explored for learning Q matrices tailored to a particular problem, although in this case we have no guarantees that the resulting Q matrix leads to optimal performance on our objective. We also think learning the Q matrix directly would be a great direction for future work, but this is difficult to do with gradient descent – optimizing through the discrete sampling step using a learned Q matrix would require some kind of Gumbel Softmax or REINFORCE-style estimator, which comes with its own set of challenges such as gradient bias or high variance.
>
> **3. Details on D3PM NN embeddings**
>
> We will update the draft to include additional details in Section A.2.4 and 5.1, including an additional figure showing the process of generating the transition matrices. For all of our experiments, we use pre-trained embeddings taken from a standard transformer language model trained on the same dataset with the same vocabulary. The adjacency matrix is constructed from the K nearest neighbors of each embedding (we tried both cosine and L2 distance).
>
> We imagine two ways one could incorporate end-to-end supervision into the learned embeddings. The first is to train the model's embeddings based only on the gradients through the transformer layers, then periodically (possibly every iteration) rebuild the transition kernel using the current embeddings. The second is to truly back-propagate through the process of constructing and sampling from the transition matrix. We ran some preliminary experiments using the first approach, but we found it computationally challenging to update the nearest neighbor graph frequently. The optimization target also becomes non-stationary and we found that it led to worse performance. We leave the second approach to future work, since both constructing the transition matrix and sampling from it are non-differentiable operations. It would be interesting to explore constructing the transition matrix from embeddings using differentiable operations (possibly based on embedding distance) and combining it with continuous relaxations of sampling (such as Gumbel-softmax) that would enable end-to-end optimization.
>
> **4. Can the auxiliary loss be seen as a reweighting of the loss terms?**
>
> The auxiliary loss is not equivalent in general to an ELBO reweighting. There may be parameterizations $\widetilde{p}\_\theta(x\_0 | x\_t)$ that are optimal under any reweighting of the ELBO but not optimal under the auxiliary loss, because the ELBO only supervises $\widetilde{p}\_\theta(x\_0 | x\_t)$ through the sum $\sum\_{x\_0} q(x\_{t-1}, x\_t | x\_0)\widetilde{p}\_\theta(x\_0 | x\_t)$.
>
> Consider the following example: suppose we have a 2-step discrete diffusion process over a sequence of length one with a vocabulary of size 4 (A, B, C, D), and let $q(x\_0)$ be a point mass distribution on A. During the first timestep, assume A transitions to B with 50% probability. During the second timestep, assume A transitions to C with 50% probability and B transitions to D with 50% probability. Without the auxiliary loss, at timestep 2 the model $\widetilde{p}\_\theta(x\_0 | x\_2)$ is free to predict a point-mass on either A or B (or a mixture of the two), either of which will have the same marginal $p(x\_1 | x\_2) = [0.5, 0.5, 0, 0]$ which exactly matches the true posterior and has $D\_{KL} = 0$. This is also optimal under any reweighting of the ELBO terms. However, with the auxiliary loss, only a point-mass on A (the true value of $x\_0$) is optimal, because we are directly supervising the quantity $\widetilde{p}\_\theta(x\_0 | x\_2)$, not just $p\_\theta(x\_1 | x\_2)$.
>
> We note that while the auxiliary loss is not in general equivalent to a reweighting, they may be equivalent in certain special cases. As one example, for absorbing diffusion each term in the KL loss is equal to a scaled version of the corresponding term in the auxiliary loss (following the argument in Appendix A.3), with the auxiliary loss giving more weight to reconstructions for larger values of $t$ (compensating for the $1 / t$ reweighting in A.3).
>
> We will add this discussion to Section 3.4 and a full proof to the appendix. We appreciate the clarification.
>
> **5. Wouldn’t we expect some diffusion model to outperform autoregressive models?**
>
> We share the intuition that there is likely some D3PM that can outperform autoregressive models, although imposing the additional constraint of non-autoregressive generation order (as we do here) makes this more difficult. We hope future work can make further progress on this using the framework we introduce. Please see our general comment for a more in-depth discussion.
>
> **6. Connections to autoregressive models have been made before in [3].**
>
> We will update Section 4 to include this reference. Our construction is more precise and actually falls within the D3PM model family (unlike the DDPM version).
>
> ---
>
> [1] Jascha Sohl-Dickstein, Eric Weiss, Niru Maheswaranathan, and Surya Ganguli. Deep unsuper395 vised learning using nonequilibrium thermodynamics. In International Conference on Machine 396 Learning, pages 2256–2265, 2015.
>
> [2] Emiel Hoogeboom, Didrik Nielsen, Priyank Jaini, Patrick Forré, and Max Welling. Argmax 362 flows and multinomial diffusion: Towards non-autoregressive language models. arXiv preprint 363 arXiv:2102.05379, 2021.
>
> [3] Jonathan Ho, Ajay Jain, and Pieter Abbeel. Denoising diffusion probabilistic models. In 360 Advances in Neural Information Processing Systems, pages 6840–6851, 2020.

---

### Author Response · Authors · 2021-08-10
**Corrections**

After submission, we identified two errors in our manuscript, which we describe here.

**Correction to Section 3.3 "Parameterization of the reverse process":** In the definition of the $x\_0$ parameterization, we mistakenly wrote $p\_{\theta}(x\_{t-1}|x\_t) = \sum\_{\widetilde{x}\_0} q(x\_{t-1} \vert x\_t, \widetilde{x}\_0)\widetilde{p}\_{\theta}(\widetilde{x}\_0|x\_t)$, whereas the actual definition we intended to use was $p\_{\theta}(x\_{t-1}|x\_t) \propto \sum\_{\widetilde{x}\_0} q(x\_{t-1}, x\_t \vert \widetilde{x}\_0)\widetilde{p}\_{\theta}(\widetilde{x}\_0|x\_t)$. This second form is necessary to ensure that we obtain the correct sparsity pattern as discussed in Section 3.3, since $q(x\_{t-1} | x\_t, \widetilde{x}\_0)$ may be undefined if $x\_t$ is not reachable from $\widetilde{x}\_0$.

**Correction to our autoregressive baseline for text8:** We uncovered an error in the post-processing logic converting nats to bits for the autoregressive baseline on the text8 dataset: instead of 1.23 bpc, the performance of this baseline is actually 1.37 bpc (lower is better). This brings the performance of the autoregressive baseline closer to that of our D3PM models, although as before the autoregressive models still have better performance.

---

> ### Public Comment · ~Clément_Vignac2 · 2022-08-08
> **Derivation of the reverse process**
>
> Dear authors,
> could you have the kindness to explain how you derived the updated parametrization of $p(x_{t-1} |x_t)$?
>
> I understand that the previous one comes from: $p(x_{t-1}|x_t) = \sum_{x_0} p(x_{t-1}, x_0 | x_t) = \sum p(x_{t-1} | x_0, x_t) p(x_0 | x_t)$, but I did not get how the term in $t-1, t | 0$ appears.
>
> I am also confused because it seems to me that $q(x_{t-1}, x_t | \tilde x_0)$ is a matrix whose rows parametrize the state and $x_{t-1}$ and the columns the state of $x_t$. Does it mean that you use a matrix multiplication?
>
> Thank you for your help!
>
> Clément

---

> > ### Public Comment · ~Jacob_Austin1 · 2022-08-09
> > **Derivation of the reverse process**
> >
> > Hi Clément, the motivation for this is a little tricky, so I hope this helps. So the first thing I'll say is that we can basically choose any parameterization we want, since it's just a question of what the NN predicts. The parameterization we found worked best is motivated by Ho et al. 2020, where they predict $x_0$ originally and then run the forward process $q$ forward to predict the noisier intermediate value.
> >
> > So the classic marginalization is just
> >
> > $$p(x_{t-1} | x_t) = \sum_{x_0} p(x_0 | x_t) q(x_{t-1} | x_t, x_0) = \sum_{x_0} \frac{p(x_0 | x_t) q(x_{t-1}, x_t | x_0)}{q(x_t | x_0)}$$
> >
> > Which is very close to what we do, but not quite (we basically drop the denominator because it's hard to compute the full sum). We have our code available on Github to view, which shows this.
> >
> > And yes, that's correct, depending on which convention you take for row and column vectors. In general, it's actually a 3D array since $x_0$ is also variable. But yes most of this computation involves a series of pointwise and matrix multiplications.

---

> > > ### Public Comment · ~Junda_Ye1 · 2023-04-04
> > > **Derivation of the discrete posterior probability (func. (3))**
> > >
> > > Dear authors, could you please give a detailed derivation of the function (3), the discrete posterior probability $q(x_{t-1}|x_t, x_0)=Cat(x_{t-1}; p=\frac{x_tQ_t^\top \odot x_0 \bar{Q}_{t-1}}{x_0\bar{Q}_t x_t^\top})$?
> > >
> > > I understand that the intermediate probabilities follow: $q(x_t|x_{t-1}, x_0) = Cat(x_t; p=x_{t-1}Q_t)$, $q(x_{t-1}|x_0)=Cat(x_{t-1}; p=x_0\bar{Q}_{t-1})$, and $q(x_t|x_0)=Cat(x_t;p=x_0\bar{Q}_t)$, but I just cannot figure out how you get the expression or the trick how you turn the three categorical distributions into one.
> > >
> > > Thanks for your help!

---

> > > > ### Public Comment · ~Christopher_Beckham1 · 2023-04-24
> > > > **Derivations**
> > > >
> > > > Junda and I had a discussion about this, and we managed to derive that expression after a little back and forth through email. I am posting that here in case it helps other folks: https://beckham.nz/2022/07/11/d3pms.html
> > > >
> > > > Thanks,
> > > > Chris

---

> > ### Public Comment · ~Najwa_Laabid1 · 2023-02-13
> > **Auxiliary loss a reweighting of L_vb?**
> >
> > Is the auxiliary loss you proposed in Eq 5 a reweighting of the variational bound loss? Why/why not? You discuss this in the second paragraph of Section 3.4. (after equation 5) but the answer is not clear to me.
> >
> > If this loss is not a KL reweighting, can it be used alone to train a D3PM diffusion model, similar to Ho(2020)'s L_simple?
> >
> > Thanks in advance!
> > Best

---

> ### Public Comment · ~Anirudh_Buvanesh1 · 2024-11-23
> **Modelling choice made in reverse process**
>
> Dear Authors,
>
> Section 3.3 in the paper makes the choice of modelling $p_{\theta}(x_{0} | x_{t})$ using a neural network rather than $p_{\theta}(x_{t-1} | x_{t})$. Could the authors please explain the rationale behind this?
>
> Intuitively, it seems that it might be easier to model $p_{\theta}(x_{t-1} \mid x_t) $ than $p_{\theta}(x_0 \mid x_t)$, as the former involves denoising a single step of forward noise, while the latter involves learning to denoise multiple steps. Hence, I'm a bit confused as to why it wouldn't be better to model $p_{\theta}(x_{t-1} \mid x_t)$?
>
> Thanks for your help!
>
> Anirudh

---

> > ### Public Comment · ~Jacob_Austin1 · 2024-11-25
> > **x_0 parameterization**
> >
> > This was originally motivated by the x_0 parameterization in https://arxiv.org/abs/2006.11239, where they find that predicting the full image rather than the noised image works better. Intuitively, I think this makes the training objective more stable, since at higher noise values the corrupted image/text is mostly mask tokens and thus not very information rich.

---

### Author Response · Authors · 2021-08-10
**Summary of reviews and responses**

We would like to thank all of the reviewers for their time and valuable feedback, and will incorporate their suggestions into our next revision. Here we give a summary of the most common points raised by the reviewers, along with our response to each.

**Clarity and technical quality:** Multiple reviewers praised the clarity of our submission, calling it "well written and easy to read" (vMWp), "technically sound" and "well written" (u96p), and "clearly written and easy to follow" (gjLx).

**Originality of the contributions:** Reviewers were somewhat divided regarding the originality of our framework and contributions. On the positive side, Reviewer u96p states that our paper "makes several nontrivial technical contributions", and that "the proposed structured transition matrices are interesting", and Reviewer vMWp states that "the idea to define transitions based on proximity in embedding space is interesting and the connections between the diffusion with absorbing states and masked language models are also insightful". On the negative side, Reviewer vMWp felt that our conceptual innovations were limited, stating that we are "merely suggesting different transition kernels for the forward discrete diffusion" and that our reconstruction loss is "not based on any novel or fundamental theory", and Reviewer gjLx describes our contributions as having "mediocre originality" and being focused on "improving practical performance, through designing effective transition matrices, loss functions, and noise schedules".

We would like to emphasize that our contributions encompass more than just the specific transition kernels, loss functions, and noise schedules mentioned by Reviewers vMWp and gjLx. Specifically, the D3PM framework itself (described in Section 3) is a significant conceptual contribution, which formalizes a large design space of possible discrete diffusion models. Prior work on discrete diffusion models ([4], [5]) only considered and derived results for diffusion models that involve Bernoulli or uniform transitions, without formalizing them in terms of Markov transition matrices. The D3PM framework in this submission goes further by showing that

- (a) any categorical Markov process can be used and that the models can be parameterized with a transition matrix Q,
- (b) different choices of Q lead to dramatically different performance, and
- (c) the same types of Markov process can perform well on both images and text.

Although the specific transition kernels we use may seem unsurprising in hindsight, our framework provides a useful lens for thinking about and comparing these models. This framework was vital for designing the embedding-space transition matrices and deriving the connections to masked language models that were praised by Reviewer vMWp, and it also opens the door to exciting future work regarding learning the Q matrices directly, as suggested by Reviewers vMWp, u96p, and uXUZ.

**Significance of language results:** As noted by Reviewers vMWp, u96p, and uXUZ, our D3PM models improve on previous diffusion-based and non-autoregressive approaches for text generation, and Reviewer u96p calls these improvements "compelling". On the other hand, all of the reviewers point out that D3PMs, like other non-autoregressive models, attain lower performance than standard autoregressive models. We agree that there is room for improvement in this domain, but emphasize that non-autoregressive text generation is a challenging problem and that our results make significant progress compared to prior work. This area is less mature, but as we have seen in speech generation [6], qualitatively new possibilities arise if significant progress can be made. Language modeling is increasingly bottlenecked by inference speed, as models get larger and generated sequences longer. Thus, it is valuable to pursue improvements even if we are not currently achieving SOTA performance.

**Significance of image results:** Reviewer gjLx states that our "image generation experiments are encouraging and can send a strong message to the research community", and Reviewer u96p states that it "achieves comparable results" to continuous denoising diffusion models. However, Reviewer vMWp summarizes our model's performance as "lower than continuous denoising diffusion models in image generation" and Reviewer ​​uXUZ notes that results are worse than recent improved denoising diffusion models such as DDPM++ and DDIM.

We note that, prior to our work, discrete-space diffusion models had not been successfully applied to standard image datasets, and even in continuous space most diffusion models use Gaussian noise without comparing it to alternative corruption processes. While our result may appear obvious in hindsight, it was not obvious that the Markov transition kernel was the key design choice for high performing models, or that any discrete diffusion model could be competitive with its continuous counterpart.

Additionally, our image generation model was based on the original DDPM model architecture and training setup [1], including the setting of all hyperparameters. Since we were able to improve negative log-likelihoods and achieve similar Inception scores and Frechet Inception Distance scores (worse than the DDPM $L_{simple}$ model but better than the DDPM $L_{vb}$ model), we hypothesize that we could (nearly) match the performance of DDPM++ and DDIM as well with discrete diffusion models by following their design choices and substituting our structured D3PM Gauss kernel.

**Significance of connections drawn:** Both Reviewer vMWp and Reviewer gjLx found the connections between our D3PM framework and previously studied autoregressive and masked language models interesting, calling them "neat" (Reviewer gjLx) and "insightful" (Reviewer vMWp). However, both reviewers also expressed concerns about their practical significance. Reviewer vMWp asks, "wouldn't we think that a more general diffusion model, as potentially defined in this paper, can outperform [BERT / masked / autoregressive models] (which is not the case)?" Reviewer gjLx states that our interpretation "doesn't give many practical insights" due to the inferior performance relative to autoregressive models.

First, we note that BERT itself is not generally viewed as a generative model, and other BERT-like models ([2], [3]) have largely either generated tokens one at a time [2] or used greedy decoding strategies that perform well for machine translation but do not correspond to probabilistic generative models [3]. In this regard, the connections we draw have already led to some practical benefits, giving us tools for building non-autoregressive unconditional generative models that incorporate masking in a principled way, and leading to strong results in the non-autoregressive setting. Our framework makes it possible to easily swap in additional corruption processes and noise schedules, which we hope leads to further progress in this area.

Second, we acknowledge that the results we present are still inferior to those of fully autoregressive models. Of course, if we took our autoregressive baseline and directly applied the reduction described in Section 4, we would obtain a diffusion model with equivalent performance, however, this would come at the cost of requiring sequential generation of tokens (as in the original autoregressive model). We share the intuition of Reviewer vMWp that there is likely to be some D3PM within our more general family that either outperforms autoregressive models, or at least admits a more flexible tradeoff between accuracy and speed. Given that we do find large differences in performance between different types of Markov transition kernel, we feel that our work takes a significant step in this direction.

**Future directions:** Multiple reviewers suggested ways in which our work could be extended or improved, including learning the transition matrices $Q_t$ end-to-end (suggested by reviewers vMWp, u96p, and uXUZ) or improving text performance using our insights in Section 4 (discussed above). We agree that these are interesting directions. Learning $Q_t$ end-to-end is nontrivial due to the non-differentiable sampling operations involved. Our D3PM NN model is a step toward this, but we leave optimizing $Q_t$ directly for future work. Likewise, we consider one possible mask-based diffusion model which leads to improved performance in the non-autoregressive setting, but leave further improvements for future work. Overall, we see these possibilities as one of the strengths of our D3PM framework: it unifies existing modeling choices and suggests a wide array of new possible generative models with their own tradeoffs and strengths.

---

[1] Jonathan Ho, Ajay Jain, and Pieter Abbeel. Denoising diffusion probabilistic models. In 360 Advances in Neural Information Processing Systems, pages 6840–6851, 2020.

[2] Wang, Alex, and Kyunghyun Cho. 2019. “BERT Has a Mouth, and It Must Speak: BERT as a Markov Random Field Language Model.” arXiv [cs.CL]. arXiv. http://arxiv.org/abs/1902.04094.

[3] Ghazvininejad, Marjan, Omer Levy, Yinhan Liu, and Luke Zettlemoyer. 2019. “Mask-Predict: Parallel Decoding of Conditional Masked Language Models.” arXiv [cs.CL]. arXiv. http://arxiv.org/abs/1904.09324.

[4] Jascha Sohl-Dickstein, Eric Weiss, Niru Maheswaranathan, and Surya Ganguli. Deep unsuper395 vised learning using nonequilibrium thermodynamics. In International Conference on Machine 396 Learning, pages 2256–2265, 2015.

[5] Emiel Hoogeboom, Didrik Nielsen, Priyank Jaini, Patrick Forré, and Max Welling. Argmax 362 flows and multinomial diffusion: Towards non-autoregressive language models. arXiv preprint 363 arXiv:2102.05379, 2021.

[6] Chen, Nanxin, Yu Zhang, Heiga Zen, Ron J. Weiss, Mohammad Norouzi, and William Chan. 2020. “WaveGrad: Estimating Gradients for Waveform Generation.” arXiv [eess.AS]. arXiv. http://arxiv.org/abs/2009.00713.

---

### Decision · Program_Chairs · 2021-09-27

**Decision:**

Accept (Poster)

**Comment:**

This paper presents a framework for learning denoising diffusion models in discrete space. Although there are valid criticisms around the novelty of the discrete denoising diffusion models and the inferior performance compared to the autoregressive models (in sampling time and NLL), the main novelty of this work is around the clever design of Markov transition matrices. Given the importance of both modeling distributions over discrete variables and text modeling without an autoregressive structure, I am recommending this paper for acceptance.